# Genome-wide CRISPR screen identifies ELP5 as a determinant of gemcitabine sensitivity in gallbladder cancer

Sunwang Xu[1,4], Ming Zhan[1,4], Cen Jiang[2,4], Min He[1], Linhua Yang[1], Hui Shen[1], Shuai Huang[1], Xince Huang [1], Ruirong Lin[1], Yongheng Shi[3], Qiang Liu[3], Wei Chen[1], Man Mohan [2] & Jian Wang [1]*

Gemcitabine is the first-line treatment for locally advanced and metastatic gallbladder cancer (GBC), but poor gemcitabine response is universal. Here, we utilize a genome-wide CRISPR screen to identify that loss of ELP5 reduces the gemcitabine-induced apoptosis in GBC cells in a P53-dependent manner through the Elongator complex and other uridine 34 ($U_{34}$) tRNA-modifying enzymes. Mechanistically, loss of ELP5 impairs the integrity and stability of the Elongator complex to abrogate wobble $U_{34}$ tRNA modification, and directly impedes the wobble $U_{34}$ modification-dependent translation of hnRNPQ mRNA, a validated P53 internal ribosomal entry site (IRES) *trans*-acting factor. Downregulated hnRNPQ is unable to drive P53 IRES-dependent translation, but rescuing a $U_{34}$ modification-independent hnRNPQ mutant could restore P53 translation and gemcitabine sensitivity in *ELP5*-depleted GBC cells. GBC patients with lower ELP5, hnRNPQ, or P53 expression have poor survival outcomes after gemcitabine chemotherapy. These results indicate that the Elongator/hnRNPQ/P53 axis controls gemcitabine sensitivity in GBC cells.

[1] Department of Biliary-Pancreatic Surgery, Renji Hospital, School of Medicine, Shanghai Jiao Tong University, Shanghai, China. [2] Department of Biochemistry and Molecular Cell Biology, Shanghai Key Laboratory for Tumor Microenvironment and Inflammation, Shanghai Jiao Tong University School of Medicine, Shanghai, China. [3] Department of Pathology, Renji Hospital, School of Medicine, Shanghai Jiao Tong University, Shanghai, China. [4]These authors contributed equally: Sunwang Xu, Ming Zhan, Cen Jiang. *email: dr_wangjian@126.com

Gallbladder cancer (GBC) is the most common and aggressive malignant cancer in biliary tract, with high mortality, poor prognosis, and a 5-year survival rate of 5–18%[1,2]. Most patients are diagnosed at the locally advanced or metastatic stage without surgical indication[3]. Chemotherapy is the major non-surgical approach for unresectable GBC patients to reduce tumor growth and inhibit tumor metastasis[3].

Gemcitabine is a nucleoside analog of deoxycytidine (2′,2′-difluoro 2′-deoxycytidine) used to treat various solid and hematologic malignant tumors, including GBC[4]. Combination gemcitabine regimens (with cisplatin or other agents) are first-line treatments for patients with locally advanced and metastatic GBC, but poor sensitivity to gemcitabine is commonly observed among GBC patients[5,6]. Although several targets and signaling pathways involved in gemcitabine resistance in GBC cells have been identified thus far[7–10], the determinant mechanism that confers gemcitabine resistance in GBC cells remains unclear. There is therefore an urgent need to uncover biological mechanisms of gemcitabine resistance in GBC and explore potentially therapeutic targets to overcome gemcitabine resistance.

RNA interference (RNAi), specifically via short hairpin RNA (shRNA) that can inactivate gene function in a sequence-specific manner, has been a predominant approach for loss-of-function screens in previous decades, but the high off-target effects and inherently incomplete protein depletion are major limitations for RNAi screens on a genome-wide scale[11]. Recently, the clustered regularly interspaced short palindrome repeat (CRISPR)-associated nuclease Cas9, along with the guidance of single-guide RNA (sgRNA), provides a novel tool to induce double-strand breaks in the DNA of both copies of the targeted genomic loci in diploid mammalian cells to create frame-shift insertion/deletion (indel) mutations, therefore achieving highly efficient and complete protein depletion and overcoming the major limitations of RNAi screen[12,13]. Genome-wide screening with CRISPR-Cas9 has dramatically enhanced the ability to perform large-scale, high efficiency loss-of-function screens both in vivo and in vitro, especially for functional gene investigations involving chemotherapeutic agent treatment[14,15].

To better comprehend the determinant mechanisms for gemcitabine resistance in GBC, we perform an unbiased genome-wide CRISPR-Cas9 loss-of-function screen in the present study. The Elongator complex subunit 5 (ELP5) has been identified as a pivotal tumor suppressor to induce gemcitabine-associated cytotoxic effects in GBC cells. We also find that uridine 34 ($U_{34}$) tRNA-modifying enzymes are essential for the gemcitabine sensitivity in GBC cells. Loss of ELP5 contributes to gemcitabine resistance by reducing the internal ribosomal entry site (IRES)-driven translation of P53 mRNA and apoptosis in a P53-dependent manner. Furthermore, we uncover the key role and mechanism of the aberrant Elongator/hnRNPQ/P53 axis in gemcitabine resistance of GBC cells and provide biomarkers to predict gemcitabine sensitivities and survival outcomes in GBC patients.

## Results

**Generation and validation of the CRISPR system in GBC cells.** To perform a stable and efficient loss-of-function screen, we applied a two-vector CRISPR system (Fig. 1a). First, we identified the NOZ cell line as the most appropriate choice for a loss-of-function screen, as it displayed the highest sensitivity to gemcitabine in a GBC cell line panel (Supplementary Fig. 1a). Then, we transduced a lentivirus carrying a *Cas9* transgene with a Flag-tag and generated a single-cell clone in NOZ cells (herein called NOZ$^{Cas9}$) (Fig. 1b). The exogenous stably expressed Cas9 did not

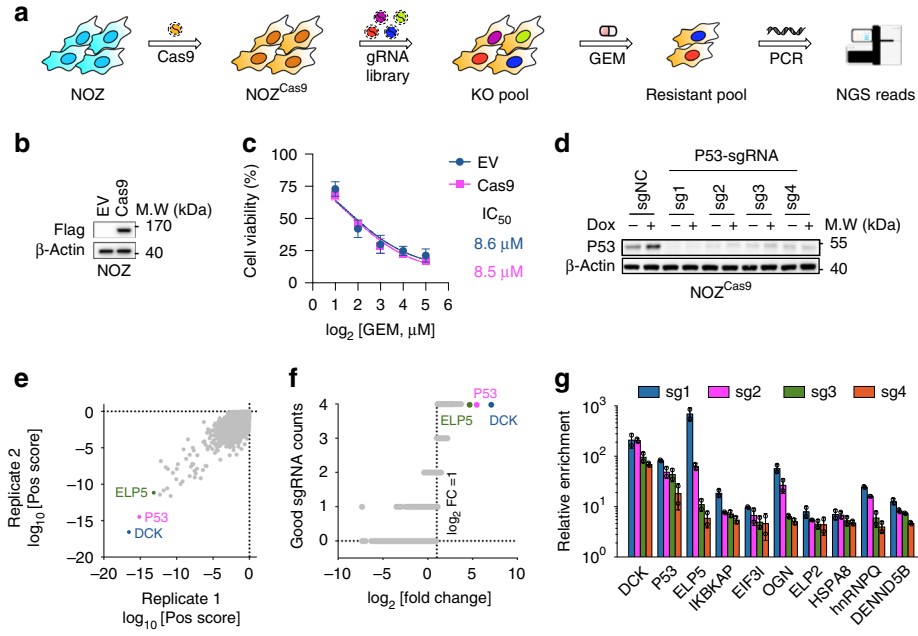

**Fig. 1** CRISPR-Cas9 genome editing efficiency and CRISPR screen results in GBC cells. **a** Schematic drawing of a positive screen for gemcitabine treatment using a two-vector system in NOZ cells. **b** A NOZ$^{Cas9}$ cell line was generated that stably expressed Flag-Cas9. **c** NOZ$^{Cas9}$ and control cells exhibit similar viability under gemcitabine (GEM) treatment at indicated doses. IC$_{50}$, 50% inhibitory concentration. **d** P53 protein was significantly depleted in NOZ$^{Cas9}$ cells infected with lentiviruses-delivered *P53*-targeting sgRNAs, followed by treatment with 1 μM doxorubicin (Dox) or vehicle for 12 h. sgNC, non-specific control sgRNA. **e–g** The sgRNA read counts after GEM treatment were normalized to the baseline counts in CRISPR screen and analyzed by MAGeCK software, and the results were presented as positive scores in two replicates (**e**), the number of good sgRNAs in two pooled replicates (**f**), and fold-changes in the top 10 genes exhibiting superiorly enriched sgRNAs (**g**). Data represent the mean ± S.D. in **c** (n = 3 independent experiments) and **g** (n = 2 independent experiments), error bars represent S.D.

impair gemcitabine sensitivity (Fig. 1c), and exhibited high knockout efficiency of the target genes at protein level (Fig. 1d).

**Loss-of-function screen in GBC cells.** We utilized an optimized genome-wide sgRNA library[16], and conducted a positive screen under gemcitabine lethal treatment in two independent replicates. After sgRNA library infection and gemcitabine treatment, the surviving cells were harvested to perform genomic DNA extraction followed by PCR to amplify sgRNA-containing regions and conduct next-generation sequencing (NGS) (Fig. 1a). The NGS reads that mapped to the sgRNA library were analyzed by MAGeCK software[17]. The positive scores were pooled for all four sgRNAs targeting specific genes, and the number of good sgRNAs targeting specific genes versus NGS read fold-changes were determined to identify essential gene hits for gemcitabine resistance (Fig. 1e–g). In all, 210 essential gene hits were identified, including deoxycytidine kinase (*DCK*), *P53*, and genes associated with metabolic pathways, DNA damage, drug resistance and apoptosis, which displayed high association with gemcitabine resistance (Supplementary Fig. 1b, c, Supplementary Data 1)[18–20]. These results demonstrate that our screening is effective to identify gemcitabine-resistant genes in GBC cells.

**ELP5 depletion induces gemcitabine resistance.** Next, we attempted to validate essential gene hits inducing gemcitabine resistance. The sgRNAs targeting the Elongator complex were significantly enriched in our screen, especially ELP5 (Elongator complex subunit 5) (Fig. 1g). Moreover, the data generated from the Cancer Cell Line Encyclopedia (CCLE) and Genomics of Drug Sensitivity in Cancer (GDSC)[21,22] showed that the ELP5 mRNA expression across over 500 cancer cell lines was negatively correlated with the 50% inhibitory concentration ($IC_{50}$) of gemcitabine (Supplementary Fig. 2a), suggesting that *ELP5* was associated with gemcitabine resistance. Therefore, we selected *ELP5* for further validation by infecting NOZ[Cas9] cells with lentiviruses containing *ELP5*-targeting sgRNAs used in our screen (Fig. 2a). The individual *ELP5*-knockout pool exhibited increased cell viability under gemcitabine treatment in a dose- and time-dependent manner (Fig. 2b, c). As a complementary approach to *ELP5* knockdown in the GBC cell lines NOZ and GBC-SD, two independent *ELP5*-shRNAs resulted in markedly increased cell viabilities under gemcitabine treatment (Supplementary Fig. 2b-e).

To obtain a complete deletion of ELP5 protein, we generated single-cell *ELP5*-knockout clones (*ELP5*[−/−]) in both NOZ and GBC-SD cell lines by CRISPR/Cas9 technique (Supplementary Fig. 3a). This approach resulted in ELP5 protein depletion at an undetectable level compared with wild-type (WT) cells transduced with non-specific control sgRNA (Fig. 2d). Cell viability, apoptosis and colony formation assays under gemcitabine treatments confirmed that *ELP5*[−/−] cells in both cell lines exhibited gemcitabine resistance (Fig. 2e–g), with minimal impairment of cell growth (Supplementary Fig. 3b, c). Resistance to cisplatin, another commonly used chemotherapeutic agent for GBC chemotherapy[5], was also observed in *ELP5*[−/−] cells (Supplementary Fig. 3d).

In xenograft models, no differences were observed in tumor volume growth and tumor weight between vehicle-treated WT and *ELP5*[−/−] tumor-bearing groups, but gemcitabine-treated *ELP5*[−/−] tumor-bearing groups exhibited markedly increased tumor volume growth and tumor weight compared with those in gemcitabine-treated WT tumor-bearing groups (Fig. 2h–j, Supplementary Fig. 3e–g). The differences in tumor proliferation and apoptosis under gemcitabine or vehicle treatment were further confirmed by KI-67 and TUNEL staining (Fig. 2k, Supplementary Fig. 3h). Together, these data demonstrate that ELP5 depletion

induces gemcitabine resistance in GBC cells both in vivo and in vitro.

**ELP5 maintains the integrity and stability of Elongator complex.** ELP5 is a subunit of the Elongator complex, which comprises two copies of each of the six subunits and is organized into two subcomplexes: the ELP123 subcomplex (ELP1, −2, and −3) possesses an acetyltransferase activity, and the ELP456 subcomplex (ELP4, −5, and −6) functions as a hexameric RecA-like ATPase to provide tRNA-specific binding sites. The Elongator complex acts as the first enzyme in the wobble $U_{34}$ tRNA modification cascade[23,24]. The wobble $U_{34}$ tRNA often harbors a 5-carbamoylmethyl ($ncm^5$) or a 5-methoxycarbonylmethyl ($mcm^5$) side chain and occasionally an additional 2-thio ($s^2$) ($mcm^5s^2$), which is required for cognate codon decoding during mRNA translation[25]. During the $U_{34}$ tRNA modification cascade, the ELP456 subcomplex hydrolyzes ATP to present a tRNA-binding site, the ELP123 subcomplex and other $U_{34}$ tRNA-modifying enzymes, including ALKBH8 and CTU1/2, sequentially catalyze the formation of 5-carbamoylmethyluridine ($cm^5U$) to $mcm^5U$ and finally $mcm^5s^2U$, respectively[23,26,27]. ELP5 is located in the ELP456 subcomplex, and directly connects ELP3 to ELP4 to unite the ELP123 and ELP456 subcomplexes and possesses an ATPase activity[23,28]. We found that loss of ELP5 resulted in the downregulated protein levels of other Elongator subunits (Fig. 3a), but not mRNA levels (Supplementary Fig. 4b); However, the expression of CTU1, CTU2, and ALKBH8 displayed no changes in protein or mRNA levels (Supplementary Fig. 4a, b) in *ELP5*-depleted cells. Further, we found that ATPase activity and thiolated ($s^2$) tRNA abundance were significantly decreased in *ELP5*-depleted cells (Fig. 3b, c). To investigate how $U_{34}$ tRNA-modifying enzymes are required for gemcitabine sensitivity in GBC cells, we generated *ELP3*, *ELP4*, *CTU1*, *CTU2*, and *ALKBH8* knockout cell individual pools, respectively. The results showed that loss of multifarious $U_{34}$ tRNA-modifying enzymes led to gemcitabine resistance (Supplementary Fig. 4c–g).

Notably, we found that under gemcitabine treatment, protein levels were increased for all Elongator subunits, although only some subunits displayed increases in mRNA levels, and CTU1, CTU2, and ALKBH8 expression remained unchanged in both protein and mRNA levels (Supplementary Fig. 4i, j). The abundance of thiolated tRNA was increased under gemcitabine treatment (Supplementary Fig. 4k). These results suggest that the $U_{34}$ tRNA modification cascade is activated under gemcitabine treatment in GBC cells.

Considering the bridging role of ELP5 in the Elongator complex, we wondered whether ELP5 depletion disrupted the integrity and stability of Elongator complex, therefore resulting in the decreased abundance of modified $U_{34}$ tRNA and gemcitabine resistance. Immunoprecipitation (IP) assays revealed that ELP5 interacted with ELP3 and ELP4 (Fig. 3d), and protein degradation assays showed that ELP5 depletion accelerated the degradation of ELP3 and ELP4 in both GBC cells (Fig. 3e). Next, we overexpressed exogenous ELP5, ELP4, and ELP3 in WT and *ELP5*[−/−] cells, respectively. In WT cells possessing the integral Elongator complex subunits, ELP5, ELP4, and ELP3 overexpression could increase the abundance of thiolated tRNA, enhance gemcitabine cytotoxic effects and increase ATPase activity (Fig. 3f–h, Supplementary Fig. 4h). However, in *ELP5*[−/−] cells displaying impaired integrity and stability of the Elongator complex, only exogenous ELP5 overexpression, which restored Elongator complex integrity, could significantly rescue the abundance of thiolated tRNA, gemcitabine sensitivity, and ATPase activity; furthermore, these were slightly rescued by either ELP4 or ELP3 overexpression when lacking endogenous ELP5 (Fig. 3f–h, Supplementary Fig. 4h). Moreover, inactivation of ATPase activity by the active site mutation within the ELP5 Asp124 residue[23], or

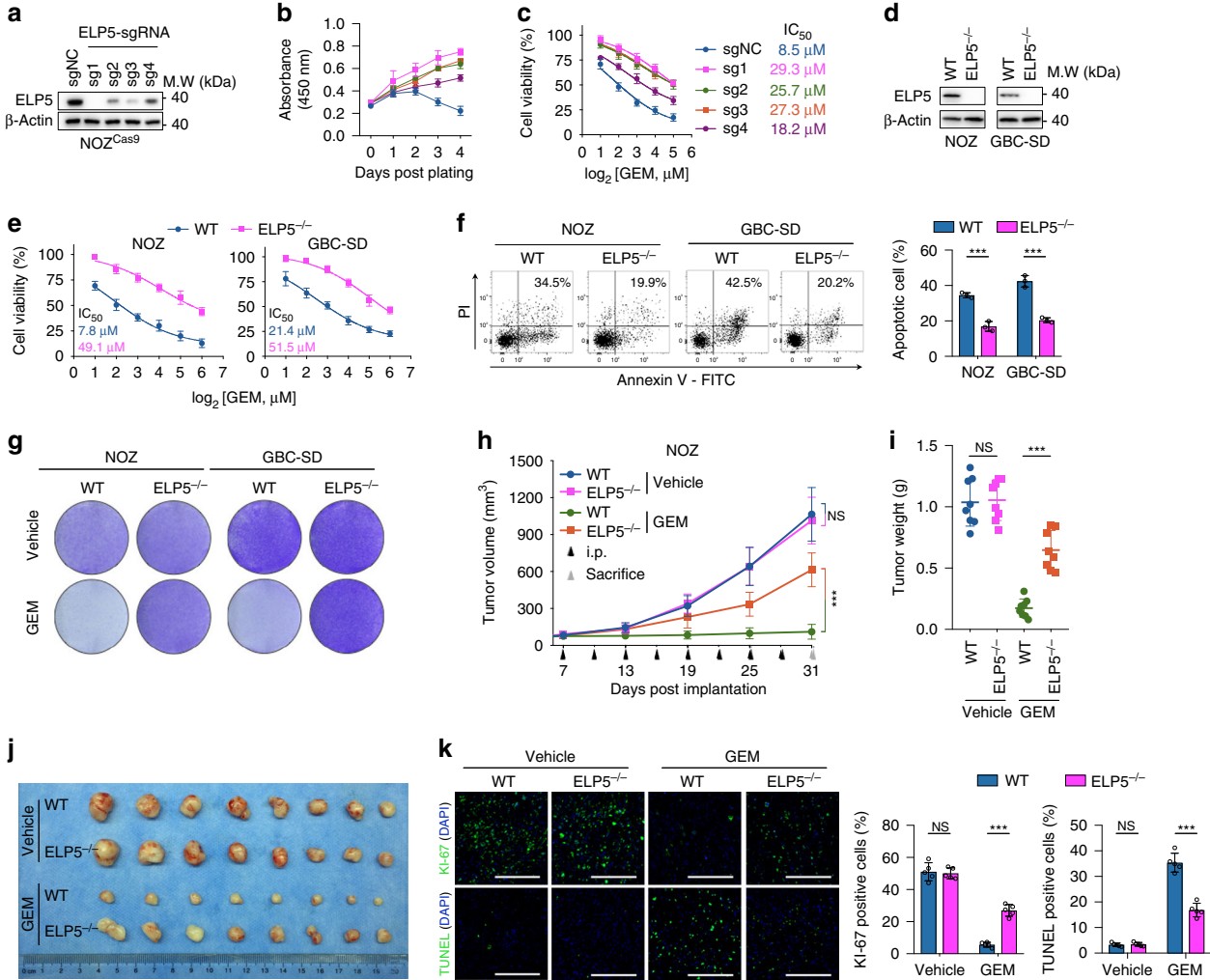

**Fig. 2** Loss of ELP5 in GBC cells mediates GEM resistance both in vivo and in vitro. **a** ELP5 protein was significantly depleted in NOZ[Cas9] cells infected with lentivirus-delivered *ELP5*-targeting sgRNAs, which were utilized in CRISPR screens. **b**, **c** *ELP5*-knockout cells exhibited increased cell viability under GEM treatment in time- (**b**) and dose-dependent manners (**c**). **d** Single-cell clones of the complete *ELP5* knockout (*ELP5*[−/−]) in two GBC cell lines, NOZ and GBC-SD, were generated using CRISPR/Cas9. The wild-type (WT) cells were transfected with the same vector carrying non-specific control sgRNA. **e**, **f** *ELP5*-depleted cells exhibited resistance to GEM, as assessed by cell viability assay under GEM treatment for 72 h at indicated doses (**e**) and apoptotic assay by Annexin-V/PI staining for flow cytometry after GEM treatment for 72 h at $IC_{50}$ (**f**). **g** Representative images of cell densities in WT and *ELP5*[−/−] cells treated with GEM at $IC_{50}$ or vehicle and stained with crystal violet. **h**–**k** ELP5 depletion prevented xenograft growth inhibition and apoptosis induced by GEM intraperitoneal injection (i.p.) in NOZ cell xenografts, but was dispensable for xenograft growth when treated with vehicle (saline), as evaluated by tumor growth volume (**h**), tumor weight (**i**), representative images (**j**) of xenograft tumors after scarification, and KI-67 (upper) and TUNEL (down) staining in paraffin-fixed xenograft tissues after scarification (**k**). Scale bars = 200 μm. $1 \times 10^6$ WT or *ELP5*[−/−] NOZ cells were injected subcutaneously into the right axilla of athymic nude mice ($n = 8$ animals per group). Data represent the mean ± S.D. in **b**, **c**, **e**, **f** ($n = 3$ independent experiments), **h**, **i** ($n = 8$ per group) and **k** ($n = 5$ per group), error bars represent S.D. Unpaired Student's *t*-tests were used in **b**, **c**, **e**, **f**, **i**, **k**, and one-way ANOVA was used in **h** (NS, non-significant, ***$P < 0.001$).

inactivation of the catalytic activity of $cm^5U$ formation by the active site mutation within ELP3 Cys109/112 residues[29] could not enhance gemcitabine sensitivity in GBC cells, but WT variants of ELP5 and ELP3 did confer enhancement (Fig. 3i–l). Taken together, these results reveal that $U_{34}$ tRNA modification is required for gemcitabine-induced cytotoxic effects in GBC cells. More importantly, ELP5 depletion leads to the disrupted integrity and stability of Elongator complex, thus interrupting its ability to drive the $U_{34}$ tRNA modification cascade.

**ELP5 is associate with drug resistance and P53 signature in GBC.** To investigate the significant altered biological signatures in ELP5-downregulated GBC, we performed Gene Set Enrichment Analysis (GSEA) of transcriptional profiles from GBC tumorous specimens (Supplementary Data 2). GSEA revealed that drug resistance signatures were significantly enriched in patients with lower *ELP5* expression compared with those with higher ELP5 (Fig. 4a, Supplementary Fig. 5a). Gene signatures representing P53-associated biological processes and oncogenes were also significantly differed between GBC patients with lower and higher ELP5 expression (Fig. 4b, Supplementary Fig. 5b). Particularly, P53-targeted genes were downregulated in GBC patients with lower ELP5 expression (Fig. 4c). RT-qPCR analysis further confirmed that transcriptional levels were indeed downregulated among a set of identified P53 target genes in *ELP5*-depleted GBC cells (Supplementary Fig. 5c). These results suggested that loss of ELP5 might silence the P53 pathway to confer gemcitabine resistance in GBC.

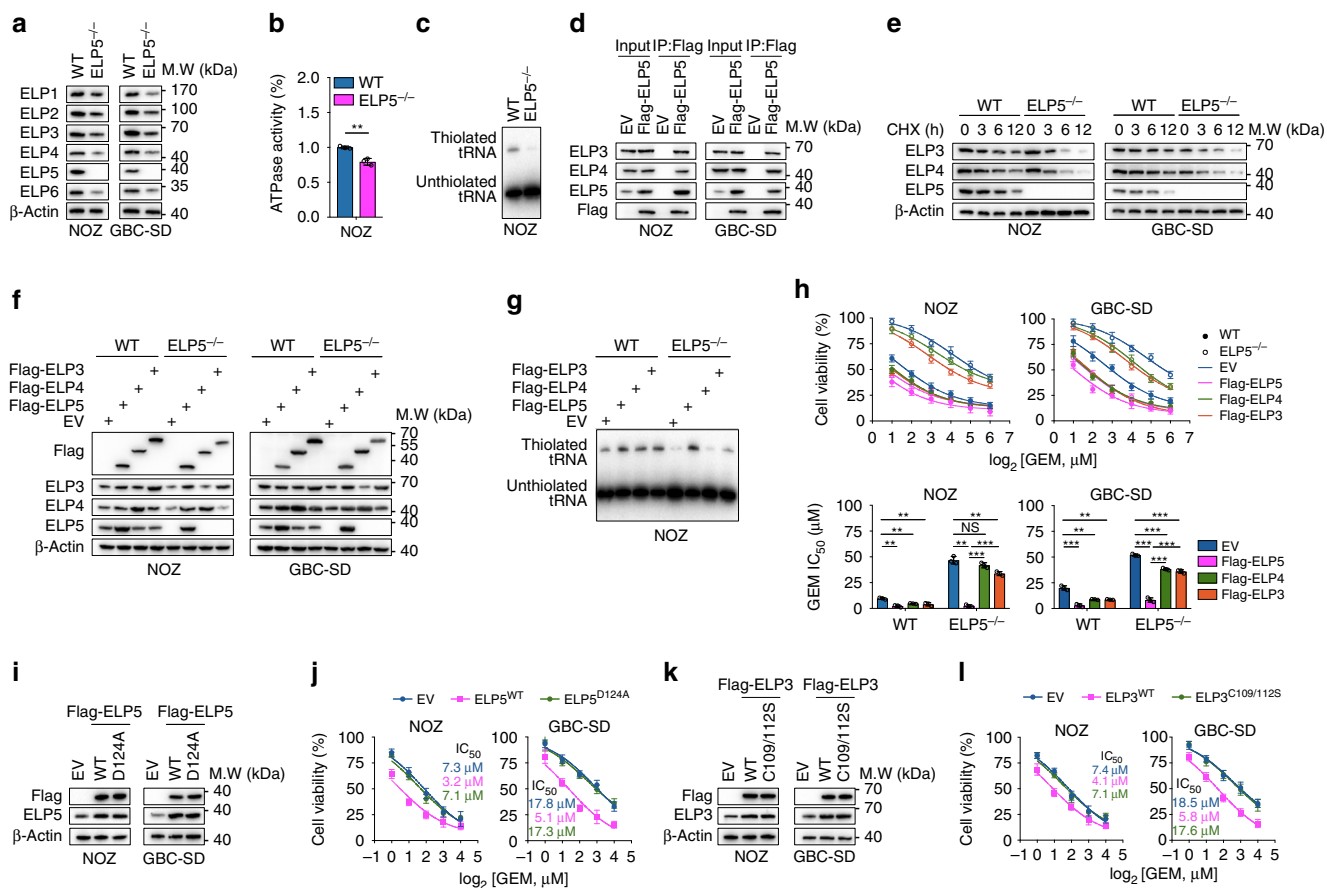

**Fig. 3** The integrity and stability of the Elongator complex are required for gemcitabine-induced cytotoxic effects in GBC cells. **a** ELP5 depletion significantly downregulated the protein levels of other Elongator subunits, including ELP1, ELP2, ELP3, ELP4, ELP6. **b** ELP5 depletion significantly decreased ATPase activity in NOZ cells. **c** The abundance of thiolated tE$^{UUC}$ tRNA was decreased in *ELP5*-depleted NOZ cells. **d** Ectopically expressed ELP5 could interact with ELP3 and ELP4 in GBC cells. **e** Endogenous ELP3 and ELP4 degraded rapidly after treatment with 50 µg ml$^{-1}$ cycloheximide (CHX) in *ELP5*-depleted GBC cells compared with WT cells. **f–h** Ectopically expressed ELP5 (**f**) could promote or rescue thiolated tRNA abundance (**g**) and GEM sensitivity (**h**) in WT and *ELP5*-depleted GBC cells, but ectopically expressed ELP3 or ELP4 (**f**) could only increase the abundance of thiolated tRNA (**g**) and GEM sensitivity (**h**) in WT cells, not in *ELP5*-depleted cells. **i, j** Mutation of the ATPase activity site residue within ELP5 (D124A) (**i**) could not promote GEM sensitivity in GBC cells (**j**). **k, l** Mutation of the cm$^5$U catalytic activity site residue with ELP3 (C109/112 S) (**k**) could not promote GEM sensitivity in GBC cells (**l**). Data represent the mean ± S.D., $n = 3$ independent experiments in **b**, **h**, **j**, **l**, error bars represent S.D. Unpaired Student's *t*-tests were used in **b**, **h** (NS, non-significant, **$P < 0.01$ and ***$P < 0.001$).

**ELP5 regulates P53 IRES-dependent translation.** To explore the hypothesis that P53 is the downstream target of ELP5, we detected P53 expression in WT and ELP5$^{-/-}$ cells with or without gemcitabine treatment. P53 and its phosphorylation at Ser46, which activates P53 to induce apoptosis[30], were both decreased in *ELP5$^{-/-}$* cells under gemcitabine treatment compared with WT cells (Fig. 5a); subsequently, reductions occurred in downstream P53 targets including P21, pro-apoptotic BAX, and cleaved Caspase-3, but anti-apoptotic BCL-2 expression was increased (Fig. 5a). Dysregulated P53 and its downstream targets were also observed under cisplatin treatment (Supplementary Fig. 6a). Furthermore, *ELP3, CTU2,* and *ALKBH8* depletion also reduced total and Ser46-phosphorylated P53 expression under gemcitabine treatment (Supplementary Fig. 6b). However, no significant changes occurred in P53 mRNA levels between WT and *ELP5$^{-/-}$* cells with or without gemcitabine treatment, but reductions in the transcriptional levels of P53 downstream targets (that is, *P21* and *MDM2*) were found in *ELP5$^{-/-}$* cells (Fig. 5b). These results indicate that U$_{34}$ tRNA-modifying enzymes are required for P53 expression and P53-mediated apoptosis, and loss of *ELP5* impairs P53 expression in a post-transcriptional manner.

To exclude the possibility that the reduced P53 in *ELP5$^{-/-}$* cells resulted from the shorter half-life of P53 protein post-translation, we evaluated the P53 protein degradation rate and determined that P53 protein degraded at the almost same rate in both WT and *ELP5$^{-/-}$* cells (Fig. 5c). This suggests that the reduction in P53 protein in *ELP5*-depleted cells does not result from the protein stability but might relate to translation efficiency.

The P53 mRNA in eukaryotic cells is canonically translated in a cap-dependent manner starting with the 5′-cap structure of m$^7$GpppN. However, under stress conditions such as apoptosis and mitosis, the alternative procedure of cap-independent translation dominates by utilizing the IRES within the 5′-untranslated region (5′-UTR) of mRNA, leading to the rapid accumulation of P53 protein[31–33]. A previous study found that tRNA-modifying enzymes could regulate IRES-dependent translation of target mRNA[34], which inspired us that ELP5 might regulate P53 IRES-dependent translation. To this end, we cloned and transfected constructs expressing the open-reading frame (ORF) of *P53* with a C-terminal Flag-tag sequence and with the IRES sequence upstream of the initiation codon (IRES-*P53*-Flag) or without (*P53*-Flag), or an empty vector (EV) in WT and

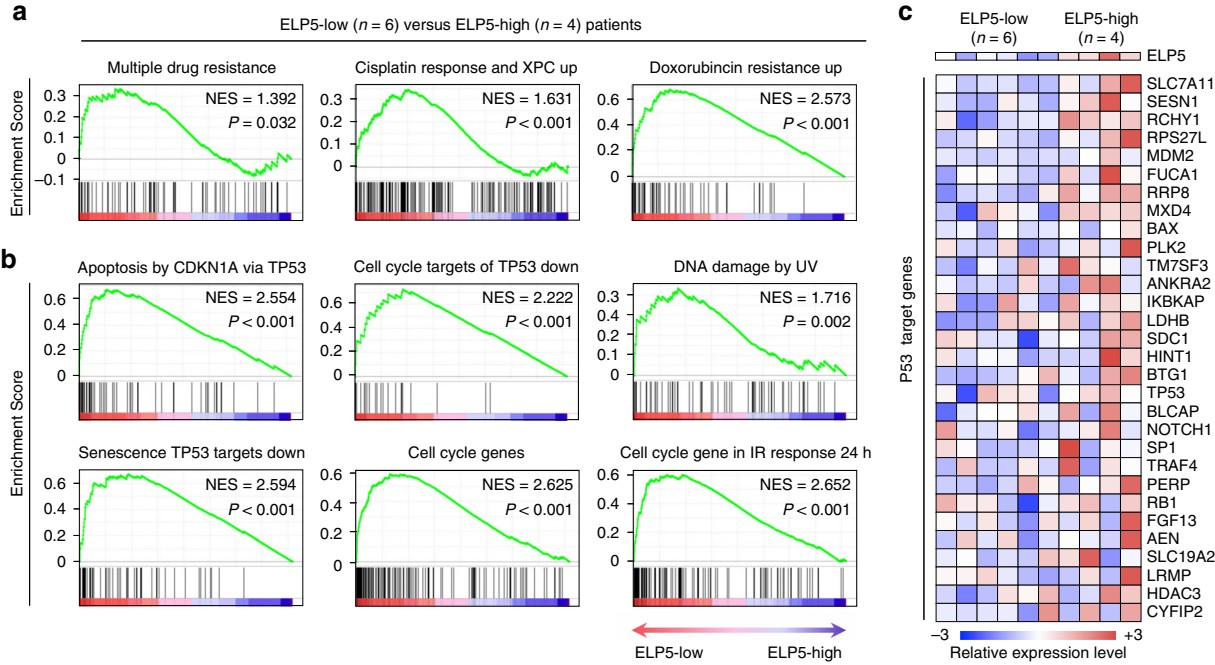

**Fig. 4** Lower ELP5 expression is correlated with drug resistance and downregulated P53 function in GBC patients. **a** Drug resistance-related biological signatures were enriched GBC specimens with low ELP5 expression. **b** Cell cycle, DNA damage, senescence, and P53-related biological signatures were dysregulated in GBC specimens with low ELP5 expression. **c** Heat map of P53 target genes that were downregulated in GBC specimens with low ELP5 expression ($n = 6$) compared to those with high ELP5 expression ($n = 4$).

$ELP5^{-/-}$ cells (Fig. 5d). Both WT and $ELP5^{-/-}$ cells expressed P53-Flag at equivalent levels (Fig. 5e), but IRES-P53-Flag expression was dramatically inhibited in $ELP5^{-/-}$ cells compared with WT cells (Fig. 5e). To further confirm that IRES-dependent translation of P53 is inhibited in ELP5-depleted cells, we generated a bicistronic construct with P53 IRES inserted between *Renilla* luciferase (Rluc) and *Firefly* luciferase (Fluc) (Fig. 5f). Rluc translation is cap-dependent, whereas Fluc translation is IRES-driven and cap-independent, and the IRES activity is calculated by the ratio of Fluc to Rluc. Remarkably, the P53 IRES activity was dramatically decreased in ELP5-depleted cells (Fig. 5g), and also decreased in ELP3-, CTU2-, and ALKBH8-depleted cells (Supplementary Fig. 6c). These results indicate that ELP5 regulates P53 expression during the initiation step of IRES-dependent translation, but not during the elongation step or protein degradation.

Finally, to validate whether ELP5 loss confers gemcitabine resistance in a P53-dependent manner, we reduced P53 expression to an undetectable level in WT and $ELP5^{-/-}$ cells. Both WT and $ELP5^{-/-}$ cells showed remarkably decreased gemcitabine-induced apoptosis without P53 (Supplementary Fig. 6d, e). Moreover, exogenous ELP5 overexpression could enhance the gemcitabine-induced apoptosis in WT GBC cells with endogenous P53, but could not rescue the gemcitabine-induced apoptosis in P53-depleted WT GBC cells (Fig. 5h, i). Taken together, these data demonstrate that, under gemcitabine-induced stress conditions, the Elongator complex and other $U_{34}$ tRNA-modifying enzymes are required to accelerate P53 accumulation by activating the initiation of IRES-driven translation to promote apoptosis in GBC cells.

**ELP5 regulates hnRNPQ translation via U34 tRNA modification.** Cellular IRES activity relies on the assistance of translation initiation factors or RNA-binding proteins that serve as IRES trans-acting factors (ITAFs) to enable the recruitment and correct positioning of the ribosome that drives the initiation of translation[31]. ELP5-depleted cells exhibited a reduced abundance of modified $U_{34}$ tRNA (Fig. 3c), and loss of modified $U_{34}$ tRNA could cause ribosomes to pause at their cognate codons and trigger the failure of protein homeostasis[35]. We speculated that ELP5 depletion might abrogate the translational efficiency of wobble $U_{34}$ modification-preferred P53 ITAFs.

Several P53 ITAFs have been identified[32,36–40]. We counted the codon content cognates to modified $U_{34}$ (i.e., $mcm^5s^2U_{34}$, $mcm^5U_{34}$, and $ncm^5U_{34}$) (Fig. 6a, Supplementary Table 1), and noted that heterogeneous nuclear ribonucleoprotein Q (hnRNPQ) was the most abundant in modified $U_{34}$ cognate codons and displayed strikingly reduced levels of protein, but not mRNA, in ELP5-depleted cells (Fig. 6b, c). Reductions in hnRNPQ protein levels, but not mRNA levels, were also observed in ELP3-, CTU2-, and ALKBH8-depleted cells (Supplementary Fig. 7a, b). hnRNPQ depletion led to reduced levels of total and Ser46-phosphorylated P53 protein, but not P53 mRNA (Fig. 6d, e), and resulted in gemcitabine resistance (Fig. 6f). However, the expression of Elongator subunits was not dysregulated by hnRNPQ depletion (Fig. 6d). P53 IRES activity was also significantly decreased in hnRNPQ-depleted cells (Supplementary Fig. 7c, d). These results confirm that reduced hnRNPQ leads to gemcitabine resistance by impairing IRES-driven translation of P53 in GBC cells.

Next, we try to confirm that hnRNPQ serves as a direct target of ELP5 in wobble $U_{34}$ tRNA modification-dependent translation. Notably, lentivirus-delivered exogenous hnRNPQ with Flag-tag could not be adequately expressed in $ELP5^{-/-}$ cells, but expressed efficiently in WT and HEK293T cells (Fig. 6g). Then, we performed ribosome IP in WT and $ELP5^{-/-}$ cells that stably expressed Flag-tagged RPL22 to quantify ribosome occupancy of hnRNPQ mRNAs. We found that ribosomes accumulated on hnRNPQ mRNAs, but not control mRNAs (i.e., β-Actin), in $ELP5^{-/-}$ cells (Fig. 6h). In $ELP5^{-/-}$ cells exogenously expressed ELP5, we observed rescued expression of hnRNPQ protein, but not mRNA, followed by rescued P53 protein expression; however,

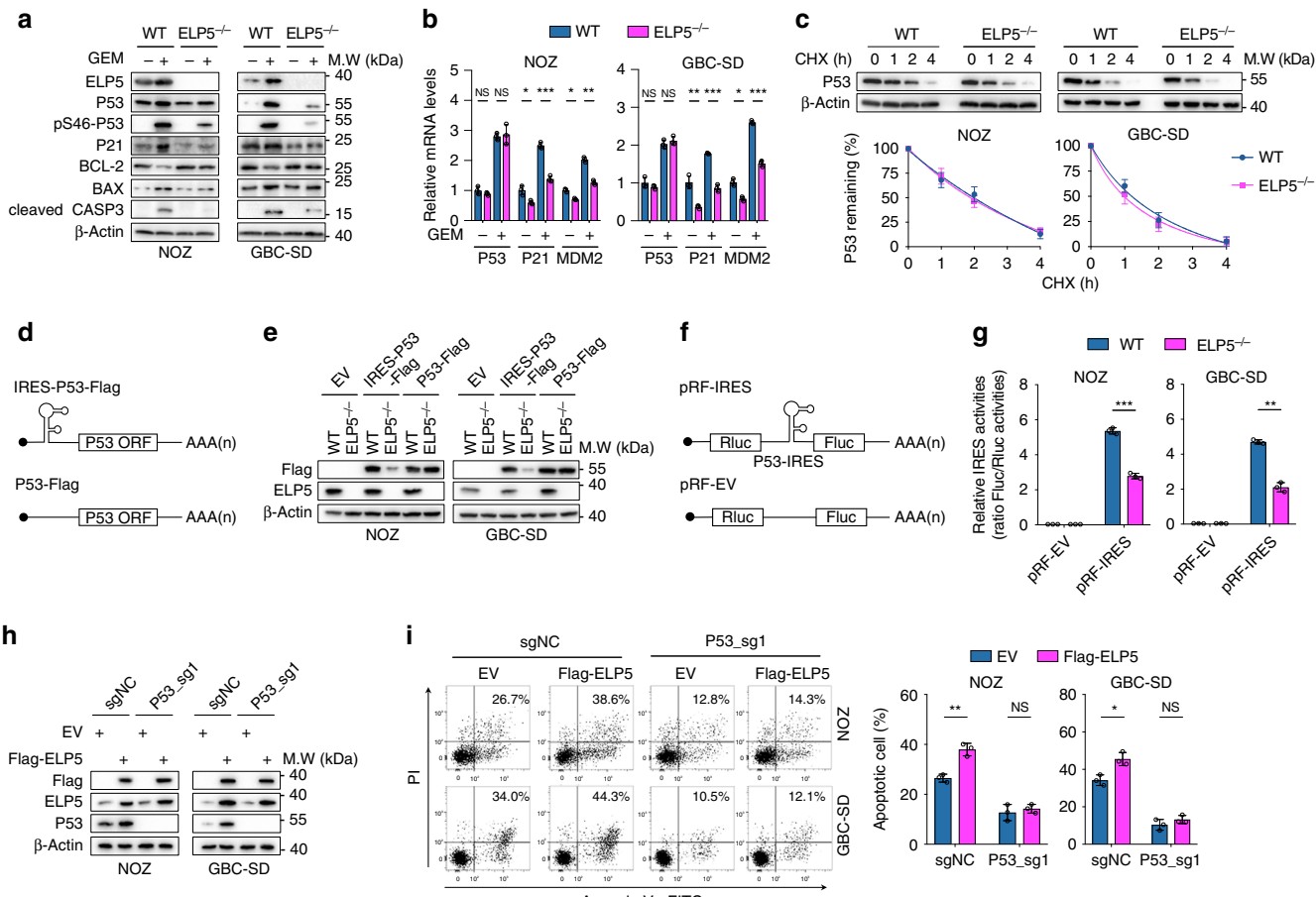

**Fig. 5** ELP5 promotes P53 expression by IRES-dependent translation. **a** ELP5 depletion substantially rescued the accumulation and activation of P53 and P53-mediated apoptosis under GEM treatment at IC$_{50}$ for 72 h in GBC cells. **b** RT-qPCR confirmed that ELP5 depletion did not affect the P53 mRNA level with or without GEM treatment, but P53 target genes (i.e., *P21* and *MDM2*) displayed dramatically reduced mRNA levels in *ELP5*$^{-/-}$ cells. **c** ELP5 depletion could not affect the P53 degradation rate in GBC cells. The P53 protein level was normalized to that of β-Actin, and the P53 protein levels in each cell were set as 1 at the 0 time point. **d** Schematic drawing of the exogenously delivered *P53* construct with or without an IRES sequence within the 5′-UTR of the *P53* ORF sequence, IRES-*P53*-Flag, and *P53*-Flag, respectively. UTR, untranslated region; ORF, open-reading frame. **e** The IRES-*P53*-Flag construct was inefficiently expressed in *ELP5*-depleted cells, but *P53*-Flag construct was adequately expressed in *ELP5*-depleted and WT cells. **f** Schematic drawing of the bicistronic construct inserted with *P53* IRES between Rluc and Fluc. Rluc, *Renilla* luciferase; Fluc, *Firefly* luciferase. **g** *P53* IRES activity was notably reduced in *ELP5*-depleted cells, as assessed by the ratio of Fluc to Rluc via dual-luciferase reporter assay. **h**, **i** Ectopically expressed ELP5 in *P53*-knockout cells (**h**) could not rescue the reduced GEM-induced apoptosis caused by P53 depletion (**i**). Data represent the mean ± S.D., n = 3 independent experiments in **b**, **c**, **g**, **i**, error bars represent S.D. Unpaired Student's *t* tests were used in **b**, **g**, **i** (NS, non-significant, *P < 0.05, **P < 0.01 and ***P < 0.001).

this was not observed in ELP3 or ELP4 exogenously expressed *ELP5*$^{-/-}$ cells with the impaired Elongator complex (Supplementary Fig. 7e, f). Notably, active site residue mutations in ELP5 and ELP3 could not facilitate hnRNPQ translation in WT GBC cells (Supplementary Fig. 7g, h). Therefore, we generated a mutant *hnRNPQ* ORF sequence in which all mcm$^5$s$^2$U$_{34}$, mcm$^5$U$_{34}$, and ncm$^5$U$_{34}$ cognate codons were replaced with synonymous codons, which did not require the modified U$_{34}$ tRNA for translation, herein called hnRNPQ$^{Um}$ (Um represents the U$_{34}$ mutation) (Fig. 6i). The hnRNPQ$^{Um}$ rescue in *hnRNPQ*-depleted cells had the ability to restore *P53* IRES activity and P53 expression comparable to that of hnRNPQ$^{WTm}$ rescue (WTm signifies that the *hnRNPQ* WT ORF sequence was synonymously mutated at sgRNA target sites) (Supplementary Fig. 8a, b). In both WT and *ELP5*$^{-/-}$ cells, lentivirus-delivered *hnRNPQ*$^{Um}$ with Flag-tag could be translated at equivalent levels (Fig. 6j) and afford comparable increases in *P53* IRES activity (Supplementary Fig. 8c). Notably, successful overexpression of hnRNPQ$^{Um}$ could rescue gemcitabine sensitivity and apoptosis in *ELP5*$^{-/-}$ cells,

nearly reaching similar levels to those in WT cells overexpressing hnRNPQ$^{Um}$ (Fig. 6k, l, Supplementary Fig. 8d). Overexpression of hnRNPQ$^{Um}$ also restored P53 translation and accumulation under both normal and stress conditions in *ELP5*-depleted cells (Fig. 6m, Supplementary Fig. 8e). The reduced P53 protein expression and gemcitabine-induced apoptosis could not be rescued by exogenous ELP5 overexpression in *hnRNPQ*-depleted cells (Supplementary Fig. 8f, g), suggesting that ELP5-facilitated P53-mediated apoptosis induced by gemcitabine was majorly by modulating hnRNPQ expression.

hnRNPQ also regulates various genes expression via cap-dependent or -independent translation[41], including *ARUKA*, *RUNX3*, *DCK*, and *PTEN*. We found that the protein levels but not mRNA levels of these genes were downregulated in *hnRNPQ*-depleted cells (Supplementary Fig. 8h, i), suggesting that these hnRNPQ targets might also be associated with gemcitabine-induced cytotoxic effects in tumor cells.

Taken together, these data demonstrate that hnRNPQ is the direct target of the Elongator complex in wobble U$_{34}$ tRNA

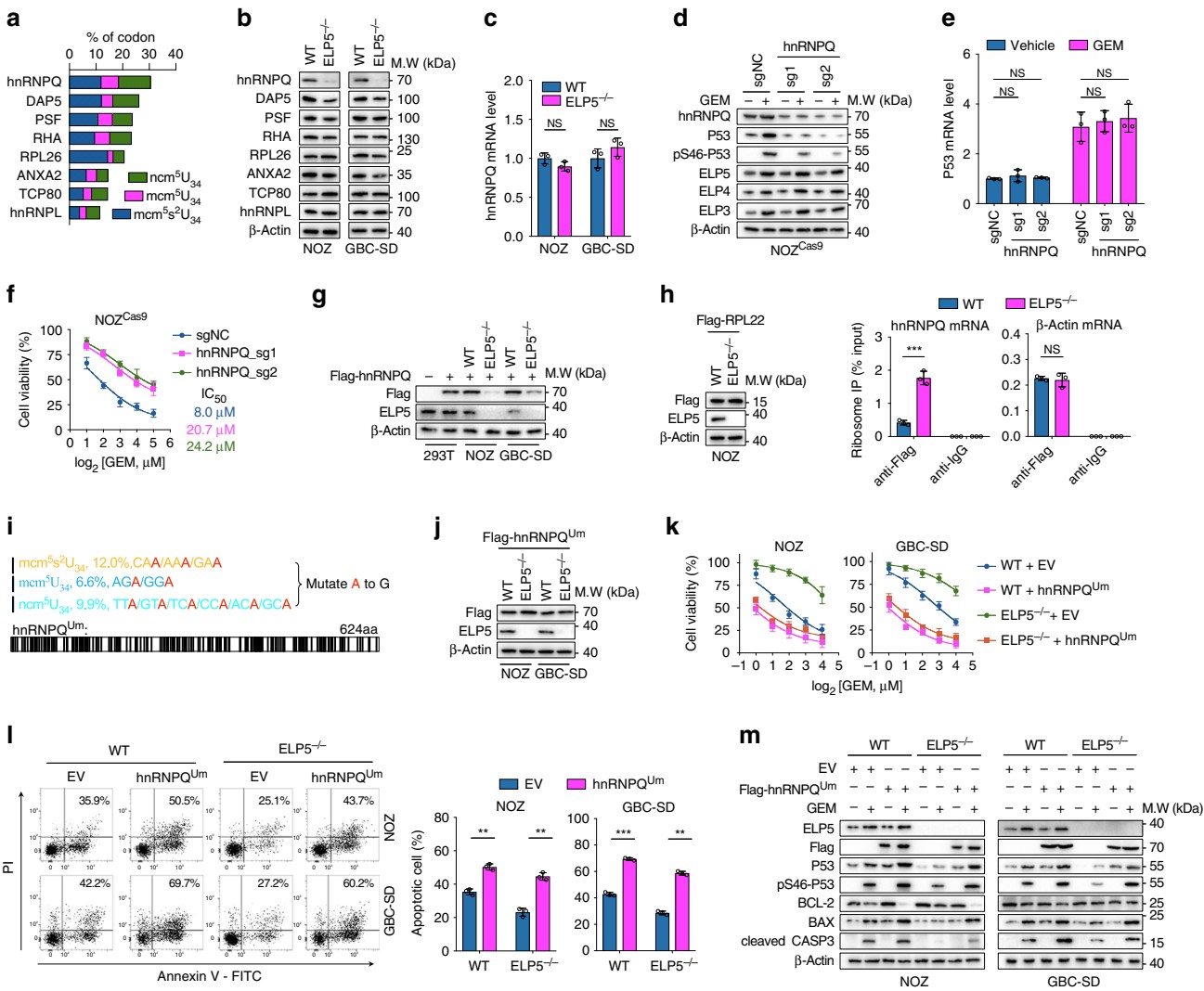

**Fig. 6** ELP5 mediates wobble U$_{34}$ tRNA modification-dependent translation of hnRNPQ. **a** The content of U$_{34}$ cognate codons of the *hnRNPQ* ORF was the most enriched in P53 ITAFs. ITAF, IRES trans-acting factor. **b, c** ELP5 depletion downregulated the expression level of hnRNPQ protein (**b**), but not mRNA (**c**). **d, e** The *hnRNPQ* knockout significantly reduced P53 protein expression and activation under GEM treatment (**d**), but no effects were observed in the expression of Elongator subunits (**d**) and P53 mRNA (**e**). **f** The *hnRNPQ* knockout exhibited GEM resistance in NOZ$^{Cas9}$ cells. **g** Exogenous hnRNPQ with a WT ORF sequence was poorly expressed in *ELP5*-depleted cells. **h** WT and *ELP5*-depleted NOZ cells stably expressed Flag-*RPL22* (left panel) to perform ribosome immunoprecipitation followed by RT-qPCR, and the results showed ribosome accumulation in hnRNPQ mRNAs in *ELP5*$^{-/-}$ cells, but not in control mRNAs (β-Actin) (right panel). **i** Schematic drawing of the wobble U$_{34}$ cognate codon mutation (Um) in the *hnRNPQ* ORF sequence (hnRNPQ$^{Um}$). **j** Flag-hnRNPQ$^{Um}$ was expressed normally in both WT and *ELP5*$^{-/-}$ cells. **k, l** Successfully overexpressed hnRNPQ$^{Um}$ could promote or rescue GEM sensitivity (**k**) and GEM-induced apoptosis (**l**) in WT and *ELP5*$^{-/-}$ cells, respectively. **m** Overexpression of hnRNPQ$^{Um}$ promoted P53 accumulation and activation in *ELP5*-depleted GBC cells. Data represent the mean ± S.D., $n = 3$ independent experiments in **c, e, f, h, k, l**, error bars represent S.D. Unpaired Student's *t* tests were used in **c, e, h, l** (NS, non-significant, **$P < 0.01$ and ***$P < 0.001$).

modification-dependent translation, and exogenous hnRNPQ overexpression could rescue P53 accumulation and gemcitabine sensitivity in Elongator complex-impaired GBC cells.

**ELP5 is associated with gemcitabine response in GBC.** To directly and quantitatively correlate associations between the Elongator/hnRNPQ/P53 axis and gemcitabine sensitivity in GBC patients, we analyzed the results of gemcitabine sensitivity in GBC mini-patient derived xenograft (mini-PDX) models in vivo (Cohort 1, Supplementary Table 2), as reported previously[42,43], and the expression of ELP5, hnRNPQ, and P53 in paraffin-fixed primary tumorous specimens. In line with the results obtained from the in vitro cell models and in vivo xenograft models above, lower level of ELP5, hnRNPQ, or P53 expression was strongly associated with

poor gemcitabine sensitivity in mini-PDX models (Fig. 7a). To further determine how the Elongator/hnRNPQ/P53 axis affects pathogenicity and survival outcome in GBC patients, we analyzed the expression of all U$_{34}$ tRNA-modifying enzymes, hnRNPQ, and P53 in a GBC tissue microarray containing 53 cases that received gemcitabine-cisplatin therapy after surgery (Cohort 2, Supplementary Table 3). GBC patients with low expression of ELP5, other U$_{34}$ tRNA-modifying enzymes, or hnRNPQ exhibited poor overall survival (OS) (Fig. 7b, Supplementary Fig. 9a). GBC patients with a mutated *P53* status exhibited poorer OS than those with WT *P53* (Supplementary Fig. 9b). Regardless of *P53* mutational status, GBC patients with low P53 expression showed poorer OS than those with high P53 expression (Fig. 7b). All U$_{34}$ tRNA-modifying enzymes, hnRNPQ, and P53 expression levels were significantly positively correlated in GBC specimens (Supplementary Fig. 9c). However,

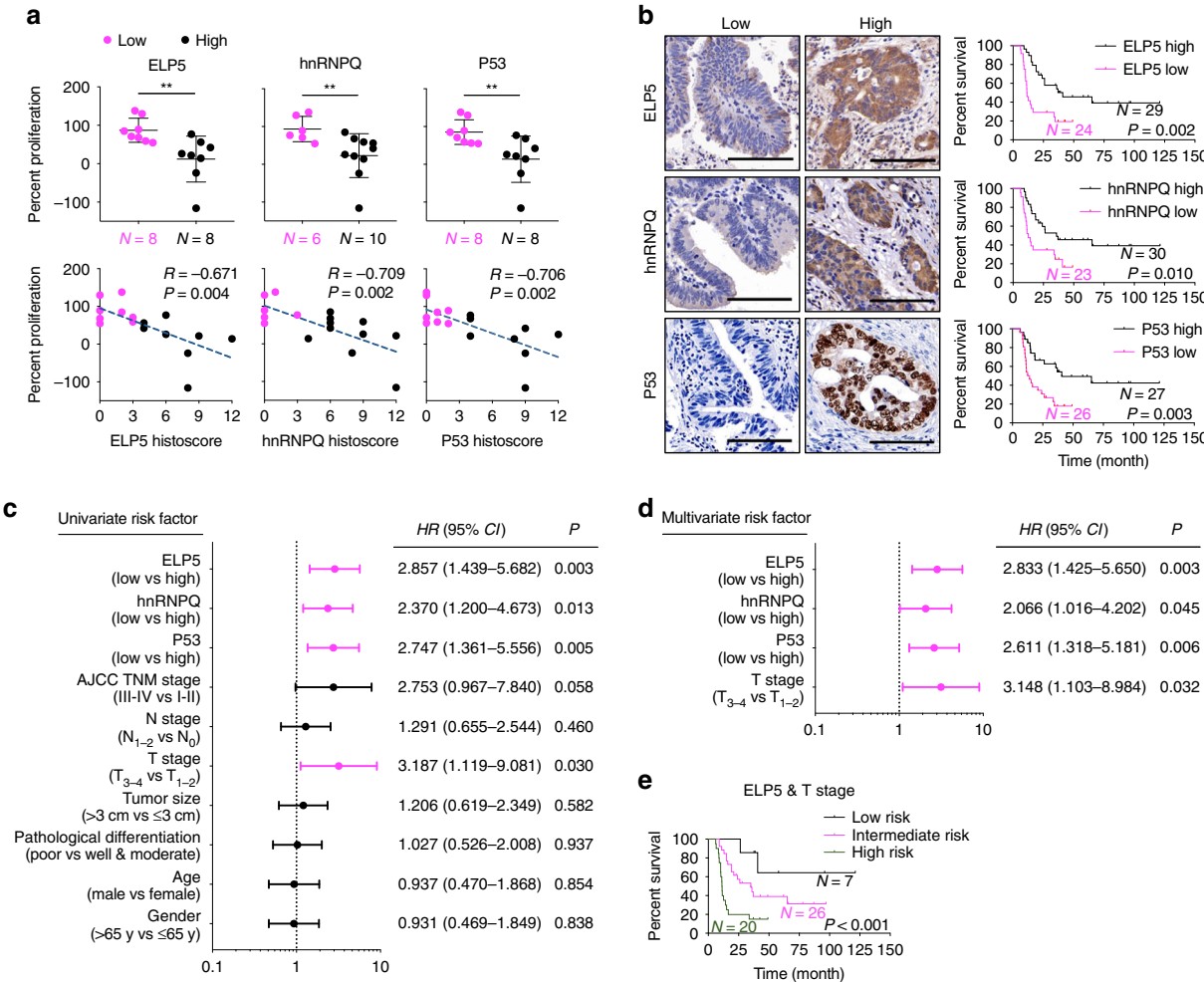

**Fig. 7** Low expression of Elongator/hnRNPQ/P53 is correlated with poor gemcitabine sensitivity and poor survival outcomes in GBC patients. **a** ELP5, hnRNPQ, and P53 expression assessed by histoscore were negatively correlated with the proliferation rate of GBC primary tumor cells under GEM treatment in mini-PDX models (Cohort 1), as normalized to vehicle treatment. Data represent the mean ± S.D, error bars represent S.D. **b** Kaplan–Meier estimate of survival time in GBC patients who received gemcitabine-cisplatin therapy after surgery (Cohort 2) according to different ELP5, hnRNPQ, and P53 levels in paraffin-fixed tumorous specimens. Scale bars = 100 μm. **c**, **d** Univariate (**c**) and multivariate (**d**) Cox regression analyses were performed in Cohort 2. The bars represent 95% CI. CI, confidence interval; HR, hazard ratio. **e** Kaplan–Meier estimate of survival time in GBC patients who received gemcitabine-cisplatin therapy after surgery (Cohort 2) based on combined ELP5 and T stages: low risk (high ELP5 and early T stage), intermediate-risk (low ELP5 and early T stage, or high ELP5 and advanced T stage), and high-risk (low ELP5 and advanced T stage) groups. Early T stage, $T_{1-2}$; advanced T stage, $T_{3-4}$. Mann–Whitney U tests were used in **a** (upper panel), the Pearson correlation coefficient were used in **a** (down panel), and log-rank tests were used in **b**, **e**.

ELP5, hnRNPQ, or P53 expression was not correlated with the other clinicopathologic features of GBC patients in Cohort 2 (Supplementary Table 4). In addition, univariate (Fig. 7c) and multivariate (Fig. 7d) Cox regression analyses demonstrated that the expression status of ELP5, hnRNPQ, or P53, or T stage was independent predictor for GBC survival with significant hazard ratios.

According to the results of multivariate analyses, we built a predictive model based on the ELP5 expression level and T stages, according to the 8th edition of the American Joint Committee on Cancer Cancer Staging Manual[44], to stratify GBC patients by their risk of poor survival under gemcitabine-cisplatin therapy. Patients in Cohort 2 were classified into the following three groups: the low-risk group (high ELP5 expression and early T stage (i.e., $T_{1-2}$)), intermediate-risk group (low ELP5 expression or advanced T stage (i.e., $T_{3-4}$)), and high-risk group (low ELP5 expression and advanced T stage). Importantly, patients in three groups displayed significantly different survival outcome risks after gemcitabine-cisplatin therapy, and the combination of low ELP5 expression and advanced T stage identified patients with a notably high-risk of poor survival (Fig. 7e). A similar predictive model was also observed in the combination of hnRNPQ or P53 expression with T stage (Supplementary Fig. 9d).

Collectively, these results further confirm that the inactivated Elongator/hnRNPQ/P53 axis is highly associated with gemcitabine therapy resistance in GBC patients, and its expression status might serve as a valuable biomarker to predict gemcitabine sensitivity and survival outcomes of GBC patients.

## Discussion

In the present study, we employ a genome-wide CRISPR screen to identify determinate genes that are essential for gemcitabine sensitivity in GBC. Here, we show that *ELP5* depletion is highly associated with gemcitabine resistance, and $U_{34}$ tRNA-modifying enzymes are critically essential for maintaining gemcitabine-induced cytotoxic effects in GBC cells. Mechanistically, loss of ELP5 leads to the abrogation of wobble $U_{34}$ tRNA modification at an early step by impairing the integrity and stability of Elongator

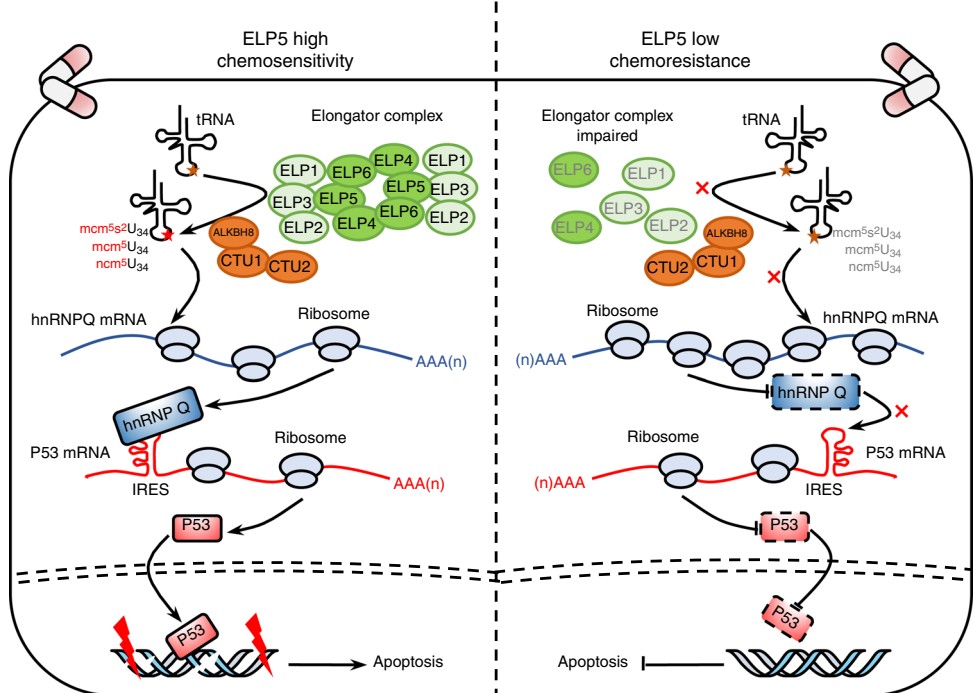

**Fig. 8** A proposed model of action by which *ELP5* depletion impairs gemcitabine-induced apoptosis in GBC cells.

complex, followed by the insufficient translation of hnRNPQ in a modified $U_{34}$ tRNA-dependent manner. As a result, insufficiently synthesized hnRNPQ protein is unable to serve as an ITAF to drive IRES-dependent translation of P53 mRNA and protein accumulation in normal and stress conditions in order to minimize potent P53-dependent apoptosis induced by gemcitabine in GBC cells (Fig. 8). We provide a critical functional connection in that hnRNPQ in Elongator modulates IRES-dependent P53 translation, which constitutes a critical signaling cascade we termed as the Elongator/hnRNPQ/P53 axis.

We support the hypothesis that *ELP5* deficiency leads to gemcitabine resistance in GBC cells, is highly associated with poor gemcitabine response in mini-PDX models, and is independently correlated with poor survival outcomes in GBC patients receiving gemcitabine-cisplatin therapy, indicating that *ELP5* expression status is highly predictive of chemotherapeutic sensitivity and survival in GBC patients receiving gemcitabine therapy. Furthermore, we have established a valuable prognostic model by combining ELP5 with T stage to efficiently stratify the risk of poor OS and whether a GBC patient will benefit from gemcitabine-cisplatin therapy after surgery, in which patients with low ELP5 and an advanced T stage possess a high-risk for poor OS, whereas high ELP5 or an early T stage indicates a moderate or low risk for poor OS.

All six subunits of the Elongator complex are required for multiple tRNA modifications in early steps of the synthesis of $mcm^5s^2U_{34}$, $mcm^5U_{34}$, and $ncm^5U_{34}$ at the wobble position of specific tRNAs[24]. The wobble $U_{34}$ tRNA modification by $U_{34}$ tRNA-modifying enzymes is critically essential for efficient translation of newly synthesized mRNAs[27]. Although the catalytic activity of $cm^5U_{34}$ is thought to reside in ELP3 of the ELP123 subcomplex, and tRNA binds the central L2 loop of ELP6 within the ELP456 subcomplex[23], it is clear that ELP5 is also important in that it is required to maintain the integrity of the Elongator complex by directly connecting ELP3 to ELP4[28]. Furthermore, the presence of ATP within the ELP456 subcomplex inhibits the interaction between the Elongator complex and tRNA,

but the ATPase activity of ELP5 can hydrolyze ATP to control tRNA binding to the ELP456 subcomplex[23]. We also confirm that ELP5 interacts with ELP3 and ELP4 in GBC cells, and ELP5 depletion reduces the expression of other Elongator subunits and accelerates ELP3 and ELP4 degradation. The consequence of these changes results in the integrity and stability of Elongator complex impaired, which is unable to drive $U_{34}$ tRNA modification cascade. A recent study demonstrated that Elongator carrying $U_{34}$ tRNA-modifying activity promotes resistance to targeted therapy in melanoma by enhancing the wobble $U_{34}$ codon-dependent translation of HIF1α mRNA and maintaining high levels of HIF1α protein[45]. However, in the present study, the depletion of $U_{34}$ tRNA-modifying enzymes could significantly reduce gemcitabine-induced apoptosis in GBC cells, thus providing the key insight that the integrity or deficiency of Elongator promotes drug resistance in a manner specific to cancer or drug types.

It is highly surprising that the expression of Elongator subunits is upregulated under gemcitabine treatment, especially the protein and mRNA expression of ELP2, ELP3, and ELP5, but this was not observed with cisplatin treatment. Although we cannot rule out the mechanism by which gemcitabine stimulates Elongator activation in the present study, we hypothesize that the inhibitory activity of gemcitabine against DNA methyltransferases[46] might demethylate the CpG island within the promoter region of Elongator subunit genes and activate transcription, as evidenced by the hypermethylation of the *ELP3* promoter in tumor tissues[47] and *ELP5* genome loci in bile duct cancer cell lines from CCLE database[21]. This hypothesis, which requires further demonstration, may provide an alternative strategy to activate the Elongator complex and enhance gemcitabine-induced apoptosis in GBC cells.

Endogenous activation or exogenous delivery of *P53* could inhibit GBC cell growth by inducing cell cycle arrest, senescence, and mitochondria-associated apoptosis[48–50]. Our data support these conclusions, as *P53* is activated by gemcitabine in GBC cells to induce apoptosis, and endogenous *P53* depletion strongly reduces gemcitabine-induced apoptosis. Previous research

showed that ELP5 is mainly located in the cytoplasm and activates the transcriptional activity of P53, but the biological mechanism of ELP5 remains unclear[51]. In line with this finding, we observed that ELP5 depletion reduces the protein level, pro-transcriptional activity and pro-apoptotic activity of P53. Dramatic accumulation and activation of P53 under stress conditions result from the cap-independent IRES-driven translation of P53 rather than cap-dependent translation or protein degradation inhibition[52]. *P53* IRES activity is highly activated following DNA damage to accelerate P53 translation[32]. However, low efficiency of IRES-driven P53 translation results in P53 inactivation, therefore increasing resistance to DNA-damaging agents and the aggressiveness of cancer cells[39,53]. Our results demonstrate that the inactivated Elongator could inhibit the *P53* IRES activity to reduce P53 accumulation and induce chemoresistance. Although *P53* is the most commonly mutated gene in GBC, with a mutational rate of 16.4–47.1% in literature[54,55] and 22.6% in present study, the majority of *P53* mutations lead to P53 inactivation and low protein levels, and over half of GBC patients retain WT *P53*. Regardless of *P53* mutational status, low P53 expression is highly associated with poor survival outcomes of GBC patients in the present study. Thus, reactivating P53 by promoting IRES-driven translation of P53, and especially retaining WT *P53*, might be a valuable therapeutic strategy for GBC.

Elongator has been reported to regulate the IRES-dependent translation of LEF1 to promote breast cancer metastasis by regulating the codon-specific translation of the ITAF protein DEK[34]. Accordingly, we find that $U_{34}$ tRNA-modifying enzymes regulate the IRES-dependent translation of P53 to maintain gemcitabine-induced cytotoxic effects through $U_{34}$ tRNA modification-dependent translation of the ITAF protein hnRNPQ, which is highly enriched in midified $U_{34}$ tRNA cognate codons in ORF sequences and significantly downregulated in GBC cells with $U_{34}$ tRNA-modifying enzymes depletion. As an RNA-binding protein, hnRNPQ regulates mRNA processing and ribosome biogenesis events, including the initiation of translation. However, the biological functions of hnRNPQ in tumors remain poorly characterized. Notably, hnRNPQ can serve as an RNA-binding protein to control myeloid leukemia stem cell programming and is required to maintain the survival of myeloid leukemia cells[56]. However, a recent study discovered that hnRNPQ, as a tumor suppressor, inhibited T-cell leukemia progression by increasing ribosomal and mitochondrial activities along with the deletion of *SNHG5*[57]. Similarly, we find that hnRNPQ acts as a tumor suppressor in GBC. Further, hnRNPQ can directly bind to the IRES region of P53 mRNA and facilitate IRES-driven P53 translation under normal and stress conditions to induce apoptosis in a P53-dependent manner[36]. Indeed, our results also confirm that hnRNPQ depletion results in the downregulation of IRES-driven P53 translation and leads to P53-dependent gemcitabine resistance in GBC cells.

hnRNPQ could regulate various genes translation aside from P53 in an IRES-dependent manner, such as *Aanat*[58], *Nr1d2*[59], and *AURKA*[41]. A previous study identified a group of genes that are potentially regulated by hnRNPQ in cap-dependent or -independent translation[41], including *RUNX3*, *DCK*, and *PTEN*, the depletion of which results in gemcitabine resistance[8,60,61]. We also found that the translation of these genes was significantly reduced in *hnRNPQ*-depleted GBC cells, but we cannot rule out the possibility that these genes may also contribute to phenotypes of gemcitabine resistance in *ELP5*-depleted cells in the present study, and this remains to be further demonstrated. Although we identify other validated P53 ITAFs in addition to hnRNPQ that are slightly altered or unchanged in *ELP5*-depleted GBC cells, whether these *P53* ITAFs also could contribute to gemcitabine resistance by regulating P53 IRES-dependent translation in GBC cells requires further examination.

In summary, we support the conclusion that the Elongator/hnRNPQ/P53 axis possesses a biological function in controlling gemcitabine-induced cytotoxic effects in GBC cells, and the dysregulation of any key process in this axis leads to an abnormal response to gemcitabine therapy in GBC. Enhancing the positive feedback in this axis may be a potential therapeutic strategy for retaining gemcitabine sensitivity in patients with GBC.

## Methods

**Patients and specimens**. Two independent cohorts of GBC patients were enrolled in this study. For the first cohort (Cohort 1, 4 males and 12 females, age 58–83 years old), as a part of the clinical trial of NCT02943031 (https://clinicaltrials.gov/), 16 cases of GBC primary tissues were utilized for gemcitabine sensitivity examination in mini-PDX models, and the related clinical information were obtained from patients. The primary tissues were obtained by radical cholecystectomy or tissue biopsy, before guided chemotherapy based on mini-PDX results, in Department of Biliary-Pancreatic Surgery, Renji Hospital affiliated to Shanghai Jiao Tong University School of Medicine between July 2016 and December 2017 with the patients' consent. Different from the previously reported[42], no guided chemotherapy results for these patients were analyzed in the present study.

For second cohort (Cohort 2, 20 males and 33 females, age 33–87 years old), a tissue microarray enrolled 53 cases of GBC tumorous tissues with complete clinicopathological and follow-up data were retrospectively obtained from GBC patients received radical cholecystectomy prior to gemcitabine based chemotherapy in Department of Biliary-Pancreatic Surgery, Renji Hospital affiliated to Shanghai Jiao Tong University School of Medicine between January 2008 and December 2014 with the patients' consent. The inclusion criteria or clinical status of patients in Cohort 2 were: (1) definitive GBC diagnosis by pathology, (2) underwent radical cholecystectomy, including complete resection of primary tumorous tissues with the negative margin confirmed by histological examination; (3) no radiotherapy or chemotherapy received before surgery; (4) after surgery, patients had received gemcitabine-cisplatin chemotherapy (gemcitabine 1000 mg m$^{-2}$ intravenous injection in 30 min, followed by cisplatin 25 mg m$^{-2}$ intravenous injection in 2 h, on days 1 and 8 every 3 weeks, at least four cycles).

The Ethics Committees of Renji Hospital affiliated to Shanghai Jiao Tong University School of Medicine approved the study protocols, and written informed consent was obtained from all subjects in this study. All the research was carried out in accordance with the provisions of the Helsinki Declaration of 1975.

**Cell culture and reagents**. NOZ cells were obtained from the Health Science Research Resources Bank (Osaka, Japan), GBC-SD, SGC-996, and EH-GB1 cells were purchased from the Cell Bank of Type Culture Collection of Chinese Academy of Sciences, and human embryonic kidney 293 T (HEK293T) cells were purchased from the American Type Culture Collection. NOZ and GBC-SD both were *P53* WT genotype ($P53^{+/+}$). GBC-SD, SGC-996, EH-GB1, and HEK293T cells were cultured in Dulbecco's modified Eagle's medium (Gibco), and NOZ cells were cultured in William's E medium (Gibco). All cell lines were supplemented with 10% fetal bovine serum (Gibco), penicillin (100 mg ml$^{-1}$) and streptomycin (100 mg ml$^{-1}$) and were incubated in a humidified chamber with 5% $CO_2$ at 37 °C. All cell cultures were ensured to be mycoplasma-negative cultures by monthly mycoplasma tests and were passaged with 0.25% trypsin containing 2.21 mᴍ ethylenediaminetetraacetic acid (EDTA) in PBS when the cells reached 80~90% confluency. Gemcitabine (GEMZAR) was purchased from Eli Lilly. Cisplatin, cycloheximide, doxorubicin, and puromycin were purchased from MedChem Express.

**Plasmids, shRNA, and sgRNA**. The ORF sequences of *ELP5*, *ELP3*, *ELP4*, *hnRNPQ*, and *P53* were cloned into the pCDH or pcDNA 3.0 vector with the Flag-tag in the N-terminus or C-terminus, and the IRES sequence of *P53* in the 5′-UTR (−134~−1) was cloned upstream of the *P53* ORF sequence. The full $U_{34}$ mutant variant of *hnRNPQ* ORF was synthesized by Biosun (Shanghai). The construct containing the *Cas9* ORF sequence (lentiCas9-Blast) was obtained from Addgene (52962). *ELP5*-targeting shRNAs and non-specific control shRNA (shNC) used in this study were obtained from Biochemistry and Molecular Cell Biology, Shanghai Jiao Tong University School of Medicine. The sgRNAs listed in Supplementary Table 5 were cloned into lentiGuide-Puro (52963, Addgene) or lentiCRISPR-V2 (52961, Addgene) in a standard protocol, and the non-specific control sgRNA (sgNC) as control.

**Virus production and infection**. HEK293T cells in 100-mm dishes were optimal for transfection at 80~90% confluency and were co-transfected with 4.44 μg of the required plasmids (overexpression constructs, shRNAs, sgRNAs, or sgRNA pooled library), 3.3 μg of psPAX and 2.2 μg of pMD2.G with 30 μl of polyethylenimine. The transfected HEK293T cells were incubated at 37 °C, and the transfection medium was replaced after 4~6 h. Virus-containing medium was collected 48~72 h

after transfection and was supplemented with 5 µg ml$^{-1}$ polybrene to infect target cells in dishes or microplates for 12~24 h. The infected cells were positively selected with 2.5 µg ml$^{-1}$ puromycin to eliminate uninfected cells to generate stable cell lines.

**Pooled CRISPR screen under gemcitabine treatment**. For the optimized Brunello sgRNA library (73178, Addgene) containing 76,441 sgRNAs targeting 19,114 protein-coding genes, with approximately four sgRNAs per gene, $8 \times 10^7$ NOZ$^{Cas9}$ cells were plated onto five 150-mm dishes to ensure sufficient coverage of sgRNAs at a low MOI (~ 0.3) to ensure that each cell was infected with < 1 sgRNA. After 24 h of infection with lentivirus containing an sgRNA library, the infected NOZ$^{Cas9}$ cells in 150-mm dishes were selected by puromycin for 5 days to eliminate uninfected cells and achieve genome-edited cell pools. Thereafter, $1.2 \times 10^7$ cells were harvested as the baseline counts for the control, and another $1.2 \times 10^7$ cells were re-plated in two 150-mm dishes, considering that each sgRNA, on average, was represented ~ 150 times (i.e., there were, on average, 150 cells infected with the same sgRNA). The re-plated cells were treated with 10 µm gemcitabine for 14 days, at a dose and time that control cells (infected with non-specific sgRNA) could not survive. Thereafter, the remaining cells were harvested, and their genomic DNA was extracted using the Blood & Cell Culture DNA Midi Kit (Qiagen), which was also extracted from the baseline count control sample. DNA fragments containing the sgRNA sequences were amplified by PCR using KOD plus DNA polymerase (Toyobo) and the primers listed in Supplementary Table 5. The PCR products of 317 bp in length containing the sgRNA sequence were subjected to NGS (Illumina HiSeq 2500) performed by a commercial vendor and analyzed by MAGeCK software[17]. The Essential gene hits were defined as genes with a read count ratio in the survival pool to the baseline control (fold change) ≥ 2, P < 0.05, and the number of good sgRNAs ≥ 3.

**Single-cell ELP5-knockout clone generation**. The modified lentiCRISPR-V2 vector, carrying Cas9 nuclease and two independently expressed sgRNAs targeting two different introns in *ELP5* genomic locus to delete exons from the second to fifth exon, containing initiation codon, was utilized to establish stable *ELP5* knockout (*ELP5$^{-/-}$*) cell lines in NOZ and GBC-SD cells, respectively. NOZ and GBC-SD cells stably transduced with non-specific control sgRNA in the same vector were defined as WT cells. Single-cell knockout clones generated and the knockout efficiency was validated by genomic PCR sequencing and immunoblotting.

**Cell proliferation assays**. Cells in single-cell suspension were plated at 4000 cells per well for chemical reagent treatment or at 2000 cells per well for growth rate assay into 96-well plates in 100 µl of culture medium, followed by assessment by the Cell Counting Kit-8 (Dojindo) assay at the indicated time points: 72 h after chemical reagent treatment or every 24 h after plating. In all, 10 µl of CCK-8 solution was added to cells directly, which were then incubated at 37 °C for 2 h, followed by measurement of the absorbance at 450 nm using a Synergy 2 microplate reader (Biotek).

**Cell apoptosis assays**. GBC cells were plated in dishes or microplates, overnight and treated gemcitabine or vehicle for the indicated time and dose. After that, all cells were collected by trypsinization without EDTA, and $1 \times 10^6$ cells were doubly stained with annexin-V-FITC/PI (BD Biosciences) and analyzed by fluorescence-activated cell sorting analysis.

**Colony formation assays**. For colony formation, cells in single-cell suspension were plated and grown in six-well plates at a density of 500 cells per well for 14 days until colonies were visible or, at a density of $1 \times 10^5$ cells per well followed by gemcitabine at concentration of IC$_{50}$ or vehicle treatment for 96 h. Later, the colonies were fixed with 4% paraformaldehyde and stained with 0.1% crystal violet.

**Western blot assays**. Immunoblotting was performed using standard procedures. Cell lysates were prepared in radioimmunoprecipitation lysis buffer (50 mM Tris, pH 7.4, 150 mM NaCl, 1% NP-40, 0.1% sodium dodecyl sulphate (SDS), 2 µM EDTA) containing proteinase inhibitor and were quantified with the Micro BCA Protein Assay Kit (Thermo Fisher Scientific). Aliquots of 20 µg of protein were electrophoresed through 10% or 15% SDS polyacrylamide gels and were then transferred to polyvinyl difluoride membranes (Millipore), followed by blocking in 5% skim milk at room temperature for 1 h and then incubation with primary antibodies at 4 °C overnight. Secondary antibodies were labeled with horseradish peroxidase, and the signals were detected using the ECL Kit (Millipore). The images were analyzed using ImageJ 1.43 software. β-Actin served as an internal control for the whole-cell lysates. Antibody against ELP5 (sc-514018, dilution 1:100) was from Santa Cruz; ELP1 (ab179437, dilution 1:5000), ELP2 (ab154643, dilution 1:1000), ELP3 (ab190907, dilution 1:3000), ELP4 (ab133687, dilution 1:1000), CTU1 (ab136083, dilution 1:500), CTU2 (ab177160, dilution 1:1000), cleaved CASP-3 (ab32042, dilution 1:500), RUNX3 (ab224641, dilution 1:1000), AURKA (ab52973, dilution 1:10000), PTEN (ab32199, dilution 1:1000) were from

Abcam; ELP6 (NBP1-91733, dilution 1:1000) was from NOVUS; P53 (2524, dilution 1:1000), pSer46-P53 (2521, dilution 1:1000) and P21 (2947, dilution 1:1000) were from Cell Signaling Technology; hnRNPQ (A7219, dilution 1:1000), ALKBH8 (A7142, dilution 1:1000) and DCK (A1794, dilution 1:3000) were from Abclonal; BAX (50599-2-Ig, dilution 1:2000), BCL-2 (12789-1-AP, dilution 1:1000), DAP5 (17728-1-AP, dilution 1:500), PSF (15585-1-AP, dilution 1:1000), RHA (17721-1-AP, dilution 1:1000), RPL26 (17619-1-AP, dilution 1:1000), ANXA2 (11256-1-AP, dilution 1:3000), TCP80 (19887-1-AP, dilution 1:1000), hnRNPL (18354-1-AP, dilution 1:500) were from Proteintech; Flag (SAB1306078, dilution 1:10000) and β-actin (A1978, dilution 1:10000) were from Sigma-Aldrich. All immunoblotting original uncropped and unprocessed images were provided in Source Data file.

**IP assays**. For IP assay, cells were transfected with Flag-ELP5 for 48 h and lysed with IP lysis buffer (50 mM Tris-HCl, pH 8.0, 150 mM NaCl, 1 mM EDTA, and 0.5% NP-40 and protease inhibitor cocktail), followed by incubated with Anti-Flag M2 affinity gel (A2220, Sigma-Aldrich) overnight at 4 °C. The immunocomplexes were subsequently washed with lysis buffer and subjected to immunoblotting.

**RNA extraction and real-time quantitative PCR (RT-qPCR)**. Total RNA was extracted from cells using TRI Reagent (Sigma-Aldrich) following the manufacturer's protocol, and 1 µg of total RNA was reverse transcribed using the PrimerScript RT Reagent Kit (Takara) into cDNA. RT-qPCR was performed in triplicate using the Applied Biosystems ViiA$^{TM}$ 7 Real-Time PCR system (Applied Biosystems). The Ct values obtained from different samples were compared using the $2^{-\Delta\Delta Ct}$ method, and β-Actin served as an internal reference gene. All primers used for RT-qPCR were listed in Supplementary Table 5.

**Northern blot assays**. In all, 10 µg of total RNA was electrophoresed through 10% polyacrylamide gels containing 0.5 × TBE, 7 m urea and 50 µg ml$^{-1}$ [(N-acryloylamino)phenyl]mercuric chloride, and transfer to nylon membrane and probed with oligonucleotide probe labeled with digoxin. The probe for tE$^{UUC}$ was listed on Supplementary Table 5.

**ATPase activity assays**. In total, $1 \times 10^6$ cells were prepared and ATPase activity was determined according to the manufacturer's protocol of ATPase/GTPase Activity Assay Kit (Sigma-Aldrich) with the 4 mM ATP solution as the standard sample.

**Luciferase assays**. For luciferase assays, cells were seed in 12-well plate at a density of $2 \times 10^5$ cells per well and incubated overnight. pRF-EV and pRF-IRES plasmid transfection were carried out using Lipofectamine 2000 (Invitrogen) according to the manufacturer's protocol. Forty-eight hours after transfection, cells were harvested, lysed and Rluc and Fluc activities were determined according to the manufacturer's protocol of Dual-Luciferase Reporter Assay System (Promega). The IRES activity was calculated by the ratio of Fluc to Rluc.

**Ribosome IP assays**. Cells were stably expressed Flag-RPL22 by lentivirus infection. After 100 µg ml$^{-1}$ cycloheximide treatment for 15 min, cells were lysed in 20 mM HEPES KOH (pH 7.3), 150 mM KCl, 10 mM MgCl$_2$, 1% NP-40, EDTA-free protease inhibitors, 0.5 mM DTT, 100 µg ml$^{-1}$ cycloheximide and 10 µl ml$^{-1}$ rRNasin and Superasin in RNase-free water. Lysates were then incubated overnight at 4 °C with agarose-beads coupling with anti-Flag or anti-IgG antibody. After incubation, beads were washed five times with a high-salt buffer (350 mM KCl) and three times with a low-salt buffer (150 mM KCl). Beads were then washed in RNA extraction buffer with β-mercaptoethanol and analyzed using RT-qPCR[45].

**Gene set enrichment analysis**. Gene expression of mRNA transcriptional profiles of GBC tumorous specimens was used to conduct GSEA to identify gene signatures between groups with low and high ELP5 expression, and the results are shown using normalized enrichment scores (NES), accounting for the size and degree to which a gene set in overrepresented at the top or bottom of the ranked list of genes with NES > 1, P < 0.05, and FDR < 0.25.

**Xenograft model**. For the xenograft experiments, 4-week-old male BALB/c athymic nude mice were housed in laminar flow cabinets under specific pathogen-free conditions with food and water provided ad libitum. In all, $1 \times 10^6$ NOZ or $2 \times 10^6$ GBC-SD WT and *ELP5$^{-/-}$* cells in 100 µl of PBS were injected subcutaneously into the right axilla of each mouse to establish the GBC xenograft model. Seven days after subcutaneous inoculation, the mice were intraperitoneal injected with vehicle (saline) or gemcitabine (50 mg kg$^{-1}$) every 3 days, with eight or five mice per group. The length and width of the tumors (in mm) were measured with calipers every 6 days before vehicle or gemcitabine injection. The tumor volume was calculated using the formula (length × width$^2$)/2. All the mice were killed at the end of the indicated intraperitoneal injection, and subcutaneous tumors were collected and weighed. The tumor volume and weight were presented as the

means ± S.D ($n = 5$–8). In vivo studies were conducted in accordance with the National Institutes of Health Guidelines for the Care and Use of Laboratory Animals, and the study procedures were approved by the Institutional Animal Care and Use Committee of Renji Hospital affiliated to Shanghai Jiao Tong University School of Medicine.

**Mini-patient derived xenograft (mini-PDX) model.** Drug sensitivity detection was carried out using the OncoVee mini-PDX assay (LIDE Biotech Inc.) following the manufacturer's instructions[42,43]. In brief, GBC tissues were obtained after surgical resection and washed with Hank's balanced salt solution (HBSS) to remove non-tumor tissue and necrotic tumor tissue in a biosafety cabinet. After cutting the tumor samples into small fragments, the fragments were incubated with collagenase solution at 37 °C for 1–2 h for digestion. The cells were collected followed by removal of the blood cells and fibroblast cells. Next, the GBC cell suspension was transferred to the HBSS-washed capsules. For subcutaneous implantation in BALB/c nude mice, a small skin incision was made, and the capsule was inserted through the subcutaneous tissue. Generally, each mouse received 3 capsules. Gemcitabine (60 mg kg$^{-1}$, intraperitoneal injection, Q4D × 2) and placebo (saline) treatments were carried out for 7 days. Finally, the capsules were removed, and the anti-tumor activity was evaluated based on the relative fluorescence units (RFU) using the CellTiter-Glo Luminescent Cell Viability Assay (Promega). The average proliferation rate of each group was calculated based on the following equation: proliferation rate = $(RFU^{D7} - RFU^{D0})_{gemcitabine}/(RFU^{D7} - RFU^{D0})_{Vehicle}$. All procedures were performed under sterile conditions at a specific pathogen-free facility and in accordance with the National Institutes of Health Guidelines for the Care and Use of Laboratory Animals. The study procedures were approved by the Institutional Animal Care and Use Committee of Renji Hospital affiliated to Shanghai Jiao Tong University School of Medicine.

**Immunohistochemistry (IHC) analysis.** The tissue slides were deparaffinized, treated with 3% $H_2O_2$ for 10 min, autoclaved in 10 mM citric sodium (pH 6.0) for 30 min to unmask antigens, rinsed in phosphate-buffered saline and then incubated with primary antibodies at 4 °C overnight, followed by incubation with biotinylated secondary antibody for 1 h at room temperature. Signal amplification and detection were performed using the DAB system according to the manufacturer's instructions, and the stained sections were photographed and converted to a digital image at 200 × under a light microscope equipped with a camera (Olympus). The intensity score was determined by evaluating staining intensity of positive staining (0 = none; 1 = weak; 2 = moderate; 3 = strong). The proportion score representing the percentage of positively stained cell (0 = none; 1 = 1–10%; 2 = 11–50%; 3 = 51–80%; 4 = 81–100%). The overall protein expression in each sample was expressed as histoscore, which was multiplication product of the intensity score (0–3) and proportion score (0–4) and is between 0 and 12, with a maximum of 12. Sample with histoscore more than four were considered to be high, and less than four were considered to be low. The staining score was evaluated by two independent pathologists. Antibody against ELP5 (HPA023279, dilution 1:200) was from Sigma-Aldrich, hnRNPQ (A7219, dilution 1:200) and ALKBH8 (A7142, dilution 1:200) were from ABclonal; P53 (GB13029-3, dilution 1:200) was from Servicebio; ELP1 (ab115223, dilution 1:200), ELP2 (ab154643, dilution 1:200), ELP3 (ab113228, dilution 1:200), ELP4 (ab133687, dilution 1:200), CTU1 (ab185473, dilution 1:50), CTU2 (ab177160, dilution 1:200) were from Abcam; ELP6 (NBP1-91733, dilution 1:200) was from NOVUS.

**Statistical analysis.** Data were presented as the means ± S.D. One-sample Kolmogorov–Smirnov test was applied for normally distributed data examination. For normal distributed data, two tailed-unpaired Student's $t$ test was applied to compare the difference between two groups, and one-way analysis of variance test was applied to compare the difference among three or more groups; for non-parametric data, Mann–Whitney $U$ test (data with abnormal distributions) was applied. For survival analysis, the Kaplan–Meier method and log-rank test were applied to determine the OS. Fisher's exact tests were applied to analyze the correlation between ELP5, hnRNPQ, P53 protein expression, and clinicopathologic features. Pearson correlation coefficient were used to analyzed the correlation of histoscore in IHC staining. All statistical calculation was performed using SPSS software package (version 23.0, IBM SPSS), and a $P < 0.05$ was considered to be statistically significant.

**Reporting summary.** Further information on research design is available in the Nature Research Reporting Summary linked to this article.

## Data availability

The authors declare that all data supporting the finding of this study are available within the paper and its supplementary information and data files. The accession number for the GBC transcriptional profiles is GSE139682. Unprocessed gel blot of Fig. 1b, d, 2a, d, 3a, c–g, i, k, 5a, c, e, h, 6b, d, g, h, j, m, Supplementary Fig. 2b, d, 4a, c, e–g, i, k, 6a, b, d, 7a, c, e–g, h, 8a, f, i, and the source data underlying Fig. 1c, g, 2b, c, e, f, h, i, k, 3b, h, j, l, 5b, c, g, i, 6c, e, g, h, k, l, Supplementary Fig. 1a, 2a, c, e, 3b–f, h, 4b, d, e–h, j, 5c, 6c, e, 7b, d, e–h, 8b–e, g, j are provided in a Source Data file.

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

## Acknowledgements

This work was supported by the grants from National Natural Science Foundation of China (81974370, 81773184, 81472240, 81272748, and 81072011 to J.W.), Shanghai Outstanding Academic Leaders Plan (2016 to J.W.), High-Level Collaborative Innovation Team Incentive Program of Shanghai Municipal Education Commission (2018 to J.W.), and Foundation of Shanghai Shen Kang Hospital Development Center (16CR2002A to J.W., and 16CR3028A to W.C.).

## Author contributions

S.X., M.Z., M.M., and J.W. participated in research design; S.X., M.Z, and C.J. conducted experiments and performed most of in vivo and in vitro experiments; M.H. and L.Y. collected the features of GBC patients, primary tissues and performed mini-PDX models. Y.S. and Q.L. performed IHC data analysis; S.X., M.Z., C.J., M.H., L.Y., H.S., S.H., X.H., R.L., and W.C. analyzed and interpreted the data; S.X. and M.Z. wrote the manuscript. All authors read and approved the final manuscript.

## Competing interests

The authors declare no competing interests.
