## [Peer Review File · Nature Communications]

Reviewers' comments:

Reviewer #1 (Remarks to the Author): Expert in gemcitabine resistance

This article presented an interesting finding linking gemcitabine resistance in GBC to protein homeostasis machinery. The results of the genetic screen indicated elongator complex protein ELP5 and p53 loss promotes gemcitabine resistance. Further, the authors found a strong link between both ELP5 and p53, hnRNPQ. ELP5 inhibition reduces hnRNPQ translation which in-turn leads to suppression of IRSE-mediated translation of p53. Overexpression of these genes made cells sensitive to GEM, suggesting a possibility of devising therapeutic interventions to overcome GEM resistance.

Overall the article is structured well, the observations of the genetic screen are supported by both in-vitro and in-vivo experiments and in the end, authors were able to convince that in GBC ELP5/hnRNPQ/p53 axis is associated with GEM sensitivity.

I have the following concerns:

1. In most of the experiments, authors have used WT parental cells as the control instead of Vector-WT controls, why? Ideally, the respective vector controls should be used.
2. What was the basis of choosing the cells used in this study? It appears from the cell viability assays performed under high GEM concentrations that the cells have some existing tolerance to GEM as some cells were still viable at the highest dose. The authors should have screened a panel of cell lines for GEM sensitivity before selecting the cells for the genetic screen.
3. The IF images in figure 1.K are poorly exposed. A higher exposure could provide a better and bright picture.

Pankaj K Singh
UNMC, Omaha

Reviewer #2 (Remarks to the Author): Expert in elongation factors

This paper describes the novel implication of ELP5, a subunit of the elongator complex, in GBC resistance to GEM. Author made a very interesting observation linking ELP5 to p53 regulation through tRNA modification.

The potential novelty of this work mostly relates to the translational regulation of p53 by tRNA modifying enzymes. However, authors fall short in the experiments provided (design and controls). The present study does not provide sufficient advance for understanding the causative link between ELP5, tRNA modification and p53 regulation. The followings need to be addressed to make a compelling case:

MAJOR CONCERNS

#1 Despite the fact that the data showing that ELP5 depletion promotes resistance to GEM in the two models are convincing, it remains unclear whether the tRNA modification activity is required in this phenotype.

- 1/ Authors should show that the loss of ELP5 leads to less (no) U34 tRNA modification (i.e. thiolation).
- 2/ Authors should also generate a loss-of-function model for other U34 tRNA modification enzymes, namely CTU1/CTU2, and/or ALKBH8.

#2 In figure 3F and G, it is very surprising that the overexpression (OE) of only one elongator subunit is able to sensitize NOZ and GBC-SD cells to GEM. Indeed, in none of the previously published papers in the field, the single unit OE could activate Elongator functions and produce a biology. Therefore, it is essential that the authors include the following controls in figure 3F-G:

- Show the expression level of other ELP subunits after OE of each of the subunits.
- Show the extent of modified tRNA (i.e. elongator substrate) after OE of each of the subunits.

- Repeat the same experiments (GEM sensitivity and read out) with catalytically inactive forms of ELP subunits (especially ELP3 and ELP5).
- Show the levels of hnRNPQ (which they show is responsible for the resistance to GEM – protein and mRNA levels) after OE of the single subunits.
- Moreover, by which mechanism is the OE of ELP subunits sensitizing the GBC cells? Would that also involve the regulation of p53 levels as shown later in the paper? Author should clarify this by providing additional experiments.

#3 Results shown in figure 4 are difficult to interpret and are confusing. Author should provide more details about each of the genes presented in the heatmap. Also, they should add tables in supplementary information for each of the GSEA analyses.

#4 Figure 5A: Author should assess the phosphorylation status of p53 in the presence or the absence of ELP5.

#5 Figure 5: Author should assess p53 levels/phosphorylation in the absence of other ELP subunits and other U34 tRNA enzymes.

#6 The experiments showing the IRES regulation of p53 (in figure 5E-F) in ELP5^{-/-} need additional controls. The construct used consists simply of the p53 5'UTR driving the expression of p53 ORF. Authors should include an experiment using a bicistronic reporter, use the p53 5'UTR IRES as the second, and to make certain there is either a long intervening RNA sequence between the two cistrons and/or increased secondary structure. These studies need to be redone, including loss of function of other U34-tRNA modification enzymes as before.

#7 Figure 6A: the levels of hnRNPQ (proteins and mRNA) should be assessed in cells lacking other ELP subunits and other U34-tRNA modification enzymes. These levels should also be detected after overexpression of ELP subunits and catalytically inactive mutants.

#8 Figure 6C: Author should include conditions treated with GEM and a western blot detecting ELP5 and other ELP subunits.

The IRES experiments (cfr point #6) should also be done in hnRNPQ loss-of-function models and after rescue with the wt, or the Um mutant.

#9 figure 6F: "We speculated that loss of ELP5 might abrogate translational efficiency of wobble U34 tRNA modification-preferred p53 ITAFs."

Author must assess the mRNA translation of hnRNPQ upon loss of ELP5.

#10 figure 6H-J: western blot showing the expression/ phosphorylation status of p53 should be added with or without GEM.

#11 figure 7. Similar analysis should be performed with other U34-tRNA modification enzymes (all ELP subunits, CTU1/2 etc). Is this only specific to ELP5?

The p53 status of the patient analyzed should be provided. The impact of p53 mutations (also seen in GBC) in this analysis should be further discussed/clarified in the manuscript.

#12 Authors must add the reference to the recent paper describing that elongator regulates the IRES-dependent translation of LEF1 through codon-specific translation regulation of the ITAF protein DEK by Delaunay et al 2016. The present work is greatly inspired by this paper (and the model herein), which is curiously not mentioned at all. This should also be added in the discussion.

#13 Line 118. It is not clear what the author wants to claim here. Can we compare GEM and CISPLATIN? Do the authors imply that cisplatin resistance occurs through a similar mechanism? If this is the case, this should be assessed experimentally.

Additional comments

#1 Figure 2K is unreadable. Image should be made more clear and a quantification should be provided.

#2 Does the expression of ELPs and U34-tRNA modification enzymes change in response to GEM?

#3 p53 is one target of hnRNPQ, it would be good to know if hnRNPQ targets expression of other genes that depend on IRES for translation such as VEGF, MYC, cIAP, and others. Authors should test the expression of other potential targets of hnRNPQ. This should also be discussed as such by the authors, who also may be interested in exploiting the possibility that besides p53 mRNA, hnRNPQ may regulate translation of a number of mRNAs encoding proteins involved in tumor resistance to GEM.

#4 In suppl table 4, other p53 ITAFs proteins appear to display a very high frequency of U34 modification codons. Therefore, they could represent ELP5 translational targets and could in one way or another contribute to the observed phenotype. Authors should test the expression of other ITAFs. This should also be further discussed as such by the authors in the manuscript.

#5 The manuscript should be strongly edited. English should be improved throughout.

Reviewer #3 (Remarks to the Author): Expert in biliary tract cancers

The authors present data from a CRISPR screen in NOZ cells treated with gemcitabine, and identify ELP5 disruption as the main driver of gemcitabine resistance. The findings are validated in one additional cell line with different probes. Biological effects of loss of ELP5 are nicely described in the manuscript. Functional interaction of ELP5 with P53 and hnRNPQ is also shown. Finally, clinical implications of ELP5 are demonstrated in human tissues and human PDX. Data are novel and comprehensive. Data are interesting and well presented. However, there are some main concerns that should be addressed, please see below.

Authors identified 210 hits that were present in the gemcitabine resistant cells. The top three were represented by DCK, P53 and ELP5. It would be useful to have information on which kind of disruption were identified in these genes from their NGS. Authors should also present the baseline mutational pattern of these cells to correctly interpret the data of the screen. Are there any P53 mutations in the NOZ cells?

The doses used for gemcitabine look very high. Even the sensitive cells (i.e. Fig 2B sg-vector control cells) have an IC50 of 10uM, which is extremely high compared to the literature where IC50 for GEM is usually in the range of nM (Lampis Gastroenterology 2018, Sekine Anticancer research 2018).

Which dose of GEM was used in the screening, and what was considered residual cells (i.e. less than 20% from original)?

Conversely, GEM seems surprisingly effective in the in vivo experiments, especially considering the low doses (50mg/kg vs 100-150 mg/kg that is usually used). Surprisingly they have quite resistant cells in vitro (IC50 in the uM range, much higher than usually used), but very sensitive xenograft models (lower doses for in vivo exp). How would authors justify this discrepancy? Did they use different forms of gemcitabine?.

Authors show that ELP5 deletion inhibits ELP3 and ELP4 proteins levels. What about the other components ELP1 ELP2 and ELP6? They should present the data.

In figure 5A: ELP5 increases under gemcitabine exposure. How do authors justify this based on the experiments where they showed that ELP5 reduction is a mechanism adopted to become resistant?

From figure 5H and 5I it looks that the main determinant of resistant is P53. Indeed reduction of P53 caused resistant also in WT, independently on ELP5. I believe these data reduces the strength of their hypothesis on ELP5 driving role, especially given that P53 resulted as a main hit from their screening.

I am not sure about the relevance of the mini-PDX, because it looks like the tumours do not grow in the mice, and are explanted after just 7 days.

Minor:

The manuscript needs an overall review of the English language, especially in the abstract and the introduction.

Page 4 line 101, please specify is it is the ELP5 mRNA expression that is associated to IC50.

Point by point response to the reviewers' comments

Reviewer #1 (Remarks to the Author): Expert in gemcitabine resistance

This article presented an interesting finding linking gemcitabine resistance in GBC to protein homeostasis machinery. The results of the genetic screen indicated elongator complex protein ELP5 and p53 loss promotes gemcitabine resistance. Further, the authors found a strong link between both ELP5 and p53, hnRNPQ. ELP5 inhibition reduces hnRNPQ translation which in-turn leads to suppression of IRSE-mediated translation of p53. Overexpression of these genes made cells sensitive to GEM, suggesting a possibility of devising therapeutic interventions to overcome GEM resistance.

Overall the article is structured well, the observations of the genetic screen are supported by both in-vitro and in-vivo experiments and in the end, authors were able to convince that in GBC ELP5/hnRNPQ/p53 axis is associated with GEM sensitivity.

--RE: We greatly appreciated your positive review.

I have the following concerns:

1. In most of the experiments, authors have used WT parental cells as the control instead of Vector-WT controls, why? Ideally, the respective vector controls should be used.

--RE: As suggested, we added new data that included vector-WT control (Figure 1b-c). We compared the gemcitabine sensitivity in vector-transfected WT NOZ cells (empty vector, EV) and NOZ cells stably expressed Cas9 (NOZ^{Cas9}). Our data confirmed that ectopically delivered Cas9 transgene did not affect GEM sensitivity or resistance in NOZ cells. The Cas9 carrying plasmid was obtained from Addgene (lentiCas9-Blast, 52962), and we deleted the Cas9-coding sequences to generate EV. We incorporated these results in the manuscript.

For other experiments, WT cells stably transfected with vector carrying non-specific control sgRNA were used as controls for ELP5 knockout cells (ELP5^{-/-}), including data in Fig. 2d-k, Fig. 3a-c & 3e-h, Fig. 5a-c, 5e & 5g, Fig. 6b-c, 6g-h & 6j-m, Supplementary Fig. 3a-h, Supplementary Fig. 4a-b, Supplementary Fig. 5c, Supplementary Fig.6a & 6d-e, Supplementary Fig.7e-f, and Supplementary Fig.8c-e. We have made corresponding corrections in the manuscript: **lines 117-118**, **Method-Single-cell ELP5 knockout clone generation (lines 546-547)** and the **Figure legend of Fig. 2d (line 850)**.

2. What was the basis of choosing the cells used in this study? It appears from the cell viability assays performed under high GEM concentrations that the cells have some existing tolerance to GEM as some cells were still viable at the highest dose. The authors should have screened a panel of cell lines for GEM sensitivity before selecting the cells for the genetic screen.

--RE: In fact, we have tested four different GBC cell lines, including NOZ, GBC-SD, EH-GB1, and SGC-996, for their cell viabilities at GEM concentration from nM to M (New Supplementary Fig. 1a). Among the four, NOZ exhibited the highest sensitivity to GEM and therefore being most appropriate for the loss of function screen like CRISPR. This is the reason why we chose NOZ. We have added this description in lines 67-69 in the revised TEXT. For GBC cell lines, a lower dose of GEM (in nM range) could not inhibit GBC cell growth. Only in μ M range would cell growth be suppressed. This is consistent with previous reports that the IC₅₀ of GEM for NOZ, GBC-SD or SGC-996 cell lines was as also in the range of μ M (Yu J, et al. *Oncol Lett*, 2018, 15:3305-3312; Wang H, et al. *Cell Death Dis*, 2017, 8:e2770; Li Y, et al. *Cell Biochem Biophys*, 2014, 70:1337-1342; Makiyama A, et al. *Anticancer Drugs*, 2009, 20:123-130).

3. The IF images in figure 1.K are poorly exposed. A higher exposure could provide a better and bright picture.

--RE: Since there was no IF or Figure 1K in figure 1, we assumed that the reviewer really meant to say the IF in Fig. 2k. We agreed with the reviewer comment. As suggested, we have now replaced the old IF images with new and higher exposure and larger magnification, and added quantificational histogram in New Fig. 2k (bar graphs on the right panel).

Reviewer #2 (Remarks to the Author): Expert in elongation factors

This paper describes the novel implication of ELP5, a subunit of the elongator complex, in GBC resistance to GEM. Author made a very interesting observation linking ELP5 to p53 regulation through tRNA modification.

The potential novelty of this work mostly relates to the translational regulation of p53 by tRNA modifying enzymes. However, authors fall short in the experiments provided (design and controls). The present study does not provide sufficient advance for understanding the causative link between ELP5, tRNA modification and p53 regulation. The followings need to be addressed to make a compelling case:

--RE: We thank the reviewer's comments and suggestions.

MAJOR CONCERNS

#1 Despite the fact that the data showing that ELP5 depletion promotes resistance to GEM in the two models are convincing, it remains unclear whether the tRNA modification activity is required in this phenotype.

1). Authors should show that the loss of ELP5 leads to less (no) U34 tRNA modification (i.e. thiolation).

--RE: We performed the Northern blot with APM-Gel to detect thiolated tRNA (with the probe for tE^{UUC}) in control and ELP5^{-/-} cells, according to *Delaunay S, et al. (J Exp Med, 2016, 213, 2503-2523)* and *Leidel S, et al (Nature, 2009, 458, 228-232)*. Briefly, we loaded 10µg total RNA to APM-Gel for Northern blot. As shown in **New Fig. 3c**, the abundance of thiolated tRNA was indeed decreased in ELP5^{-/-} cells. Moreover, overexpression of ELP5 in control cells greatly enhanced the levels of thiolated tRNA (**New Fig. 3g, lanes 1-2**).

2). Authors should also generate a loss-of-function model for other U34 tRNA modification enzymes, namely CTU1/CTU2, and/or ALKBH8.

--RE: As suggested, we knocked out (KO) CTU1, CTU2, and ALKBH8, respectively, by CRISPR/Cas9 technique with two different sgRNAs in NOZ cells. As shown in **New Supplementary Fig. 4e** (CTU1 KO), **New Supplementary Fig. 4f** (CTU2 KO) & **New Supplementary Fig. 4g** (ALKBH8 KO), loss of CTU1, CTU2, and ALKBH8 in GBC cells also contributed to GEM resistance.

#2 In figure 3F and G, it is very surprising that the overexpression (OE) of only one elongator subunit is able to sensitize NOZ and GBC-SD cells to GEM. Indeed, in none of the previously published papers in the field, the single unit OE could activate Elongator functions and produce a biology. Therefore, it is essential that the authors include the following controls in figure 3F-G:

1). Show the expression level of other ELP subunits after OE of each of the subunits

--RE: As suggested, we overexpressed ELP3, ELP4, or ELP5, respectively, in the control and ELP5^{-/-} cells (**Fig. 3f**). We found that the overexpression of ELP5 in ELP5-depleted cells could rescue ELP3 and ELP4 expression. However, overexpression of either ELP3 or ELP4 in either the control or ELP5-depleted cells did not affect the expression of any other subunit.

2). *The extent of modified tRNA (i.e. elongator substrate) after OE of each of the subunits*

--RE: As suggested, we performed Northern blot with APM-Gel and detected the abundance of thiolated tRNA (tE^{UUC}) after OE of ELP3, ELP4, and ELP5 in control and ELP5^{-/-} cells (New Fig. 3g). We found that overexpression of ELP3, ELP4, or ELP5, respectively in control cells, which contains the integrated structure of elongator complex, could activate Elongator complex and increase the abundance of thiolated tRNA; In ELP5-depleted cells, while overexpressing ELP3 and ELP4 were unable to rescue the abundance of thiolated tRNA, overexpressing ELP5 could effectively rescue the abundance of thiolated tRNA.

Considering *a*) ELP5 directly connects ELP3 of the ELP123 subcomplex to ELP4 of the ELP456 subcomplex (Close P, et al. *J Biol Chem*, 2012, 287, 32535-32545.), *b*) we found ELP5 interacted with ELP3 and ELP4 (Fig. 3d), and loss of ELP5 could accelerate ELP3 and ELP4 degradation in GBC cells (Fig. 3e), and *c*) ELP5 provides an ATPase enzymatic activity to hydrolyze ATP to present a tRNA binding site in ELP456 subcomplex (Glatt S, et al. *Nat Struct Mol Biol*, 2012 19, 314-320.), we speculate that in ELP5-depleted GBC cells, the stability of the elongator complex is impaired and its structure is disintegrated, which led to the inactivation of the elongator complex and subsequent constraint of U34 tRNA modification cascade. We have added this in the Discussion in the TEXT (lines 384-397).

3). *Repeat the same experiments (GEM sensitivity and read out) with catalytically inactive forms of ELP subunits (especially ELP3 and ELP5).*

--RE: ELP5 has an ATPase enzymatic activity in amino acids of E61 and D107 (in yeast) (Glatt S, et al. *Nat Struct Mol Biol*, 2012, 19, 314-320). We blast-searched yeast protein sequence with human sequence, found that only D107 (in yeast) is conserved in human (D124), thus we generated catalytically inactive ELP5^{D124A} mutant, as well as ELP5^{WT} and empty control vector (EV) (New Fig. 3i). For ELP3, the catalytic activity is resided in C95/98 in yeast, corresponding to the C109/112 in human (Selvadurai K, et al. *Nat Chem Biol*, 2014, 10, 810-812.). Thus we generated catalytically inactive ELP3^{C109/112S} double-mutant, as well as ELP3^{WT} and empty control vector (EV) (New Fig. 3k), These newly created vectors were individually transfected into two GBC cell lines, NOZ and GBC-SD (New Fig. 3i & 3k), and GEM sensitivity assays were performed (New Fig. 3j & 3l). Our results demonstrated that only

ELP5^{WT} or ELP3^{WT}, but not the catalytically inactive forms (ELP5^{D124A} or ELP3^{C109/112S}), could enhance GEM sensitivity in GBC cells (New Fig. 3j & 3l).

4). Show the levels of hnRNPQ (which they show is responsible for the resistance to GEM -protein and mRNA levels) after OE of the single subunits.

--RE: As suggested, we now showed the expression levels of hnRNPQ protein and mRNA in New Supplementary Fig. 7e & 7f (e, for NOZ cells and f, for GBC-SD cells; left panel for protein levels and right two panels for mRNA levels). Overexpressing ELP3, ELP4, and ELP5 in WT cells could up-regulate hnRNPQ at the protein levels, but not at the mRNA levels. However, in ELP5-depleted cells (ELP5^{-/-}), only the overexpression of ELP5, but not ELP3 or ELP4, could resume the hnRNPQ protein expression. These data further support our hypothesis that loss of ELP5 impaired the integrity and stability of Elongator complex to abrogate U34 modification.

5). Moreover, by which mechanism is the OE of ELP subunits sensitizing the GBC cells? Would that also involve the regulation of p53 levels as shown later in the paper? Author should clarify this by providing additional experiments.

--RE: To clarify this, we have now examined the expression of P53 at the levels of both protein and mRNA levels after overexpressing ELP3, ELP4, and ELP5, separately, in WT and ELP5^{-/-} GBC cells respectively. As shown in New Supplementary Fig. 7e & 7f, overexpressing ELP3, ELP4, and ELP5 in WT cells did enhance P53 expression at protein levels but not at mRNA levels. In ELP5^{-/-} cells, however, overexpression of only ELP5, but not ELP3 or ELP4, could rescue the P53 protein levels, similar to the expression patterns of hnRNPQ (New Supplementary Fig. 7e & 7f) and GEM sensitivity (in Fig. 3h). Therefore, we conclude that overexpression of Elongator subunits sensitizes GBC cells via regulating P53 expression at the protein levels.

#3 Results shown in figure 4 are difficult to interpret and are confusing. Author should provide more details about the each of the gene presented in the heatmap. Also, they should add tables in supplementary information for each of the GSEA analysis.

--RE: For a better understanding of the GSEA results in Fig. 4, we have revised this Figure as the reviewer suggested:

1) We moved the GSEA-drug resistance signatures (Fig. 4c in the initial version) to Fig. 4a in the revision, while retaining Fig. 4b for GSEA-P53 related signatures.

2) The heatmap in Fig. 4a in the initial version represents the top 50 (out of total 100) up- or down-regulated genes in ELP5-low and ELP5-high group. To easily interpret the heatmap and GSEA results for P53 related signatures, in the revision we removed the heatmap of Fig. 4a in initial version, and added a new heatmap as **New Fig. 4c**, which represents P53 target genes significantly differently expressed between ELP5-low and ELP5-high group.

3) We performed a RT-qPCR analysis showing a group of validated P53 target genes was downregulated in ELP5 depleted GBC cells (**New Supplementary Fig. 5c**).

4) We added a new table (**New Supplementary Table 4**) for each of the GSEA analysis results in the **New Supplementary**.

#4 Figure 5A: Author should assess the phosphorylation status of p53 in the presence or the absence of ELP5.

--RE: As suggested, in the revised **Fig. 5a**, we added the phosphorylation status of P53 at Ser46, the phosphorylation of which was known to activate P53, leading P53-mediated apoptosis (*Oda K, et al. Cell, 2000, 102, 849-862.*). Our data demonstrated that, under gemcitabine treatment, the phosphorylation of Ser46 was indeed decreased in ELP5-depleted cells when compared with the WT cells (**Fig. 5a**). Furthermore, this decrease in Ser46 phosphorylation occurred concomitantly with the decrease in total P53 expression (**Fig. 5a**). Therefore, we conclude that, under gemcitabine treatment, the levels of both phosphorylated (at Ser46) and total P53 were down regulated in ELP5^{-/-} cells.

#5 Figure 5: Author should assess p53 levels/phosphorylation in the absence of other ELP subunits and other U34 tRNA enzymes.

--RE: As suggested, we added new supplementary results showing the P53 levels/phosphorylation status in the absence of ELP3, CTU2, and ALKBH8 in NOZ^{Cas9} cells under GEM treatment (**New Supplementary Fig. 6b**). Our new data showed that loss of ELP3, CTU2, and ALKBH8 reduced the levels of both total protein expression and phosphorylation (activation) of P53.

#6 The experiments showing the IRES regulation of p53 (in figure 5E-F) in ELP5^{-/-} need additional controls. The construct used consist simply of the p53 5'UTR driving the expression of p53 ORF. Authors should include an experiment using a bicistronic reporter, use the p53 5'UTR IRES as the second, and to make certain there is either a long intervening RNA sequence between the two cistrons and/or increased secondary structure. These studies need to be redone, including loss of function of other U34-tRNA modification enzymes as before.

--RE: As suggested, we generated a bicistronic construct with P53 IRES inserted between Renilla luciferase (Rluc) and Firefly luciferase (Fluc) (schematic drawing in **New Fig. 5f**) according to published paper (Kim DY, et al. *Cell Death Differ*, 2013, 20, 226-234; Seo JY, et al. *Oncotarget*, 2017, 8, 51108-51122). We transfected pRF-IRES and pRF-EV in WT and ELP5^{-/-} cells, respectively, and examined the P53 IRES activity by calculating the ratio of Fluc to Rluc via Dual-Luciferase Reporter Assay System. As shown in **New Fig. 5g**, the P53 IRES activity was significantly decreased in ELP5^{-/-} cells when compared to that in the WT cells, confirmed that loss of ELP5 inhibits P53 IRES-dependent translation.

We also assessed the P53 IRES activity in ELP3-, CTU2- and ALKBH8-depleted NOZ^{Cas9} cells, respectively. As shown in **New Supplementary Fig. 6c**, loss of ELP3, CTU2, and ALKBH8 in NOZ^{Cas9} cells also inhibit P53 IRES activity.

#7 Figure 6A: the levels of hnRNPQ (proteins and mRNA) should be assessed in cells lacking other ELP subunits and other U34-tRNA modification enzymes. These levels should also be detected after overexpression of ELP subunits and catalytically inactive mutants.

--RE: As suggested, we assessed the expression level of hnRNPQ protein and mRNA in ELP3-, CTU2- and ALKBH8-depleted NOZ^{Cas9} cells. As shown in **New Supplementary Fig. 7a & 7b** (a, for protein and b, for mRNA), loss of either ELP3, CTU2, or ALKBH8 down-regulate hnRNPQ protein expression without affecting its mRNA expression.

We also checked hnRNPQ protein and mRNA expression in the GBC cells overexpressing ELP5^{WT}, ELP5^{D124A}, ELP3^{WT}, and ELP3^{C109/112S}, respectively. As shown in **New Supplementary Fig. 7g-h** (g, overexpression of ELP5^{WT} & ELP5^{D124A}; h, overexpression of ELP3^{WT} & ELP3^{C109/112S}), overexpressing ELP5^{WT} and ELP3^{WT} increased hnRNPQ at the protein levels but not at its mRNA levels. However, overexpressing ELP5^{D124A} and ELP3^{C109/112S} did not affect the hnRNPQ expression at the levels of either protein or mRNA.

#8 Figure 6C: Author should include conditions treated with GEM and a western blot detecting ELP5 and other ELP subunits.

--RE: As suggested, we added the results of new experiments with GEM treatment in hnRNPQ knockout and control cells, and showed the protein expression of P53, Ser46-phosphorylated P53, ELP5, ELP4, and ELP3 in the revised version in **New Fig. 6d**, as well as P53 mRNA levels in **New Fig. 6e**. We found hnRNPQ knockout down-regulated the expression of P53 and Ser46-phosphorylated P53 under GEM treatment without affecting the expression of ELP5, ELP4, and ELP3 protein, or P53 mRNA.

The IRES experiments (cfr point#6) should also be done in hnRNPQ loss-of-functions models and after rescue with the wt, or the Um mutant.

--RE: As shown in **New Supplementary Fig. 7d**, P53 IRES activity assessed by Fluc/Rluc Dual-Luciferase Reporter Assay System was also significantly inhibited in hnRNPQ knockout cells.

We then overexpressed hnRNPQ WTm and Um mutant in hnRNPQ knockout and control NOZ cells and performed IRES experiments (Results were shown in **New Supplementary Fig. 8a & 8b**). Because of hnRNPQ knockout cells were stably transfected hnRNPQ-targeting sgRNA, the hnRNPQ wild-type (WT) overexpression plasmid was synonymously mutated at sgRNA target site and labeled as WTm in **New Supplementary Fig. 8a & 8b**. The results showed that overexpressing hnRNPQ WTm and Um mutant could promote P53 expression and IRES activity in hnRNPQ knockout cells.

We also assessed the P53 IRES activity in ELP5^{-/-} and WT cells transfected with hnRNPQ Um mutant. As shown in **New Supplementary Fig. 8c**, overexpressing hnRNPQ Um mutant in ELP5^{-/-} restored the levels of P53 IRES activity to similar levels as they were in WT cells transfected with hnRNPQ Um mutant.

#9 figure 6F: “We speculated that loss of ELP5 might abrogated translational efficiency of wobble U34 tRNA modification-preferred p53 ITAFs.” Author must assess the mRNA translation of hnRNPQ upon loss of ELP5.

--RE: We performed ribosome immunoprecipitation and qRT-PCR in WT and ELP5^{-/-} NOZ cells, which stably expressing Flag-RPL22 following the protocol published by Rapino F, et al. (*Nature*,

2018, 558, 605-609). As shown in **New Fig. 6h**, our data demonstrated that in ELP5^{-/-} NOZ cells, hnRNPQ mRNAs, but not the β -Actin mRNA (control), were significantly enriched in ribosomes. This data indicates the hnRNPQ mRNA translation is abrogated in the absence of ELP5, consistent with (Rapino F, et al. Nature, 2018, 558, 605-609).

#10 figure 6H-J: western blot showing the expression/ phosphorylation status of p53 should be added with or without GEM.

--RE: Figure 6H-J in initial version represented Flag-hnRNPQ Um mutant expression in WT and ELP5^{-/-} cells, the cell viability and calculated GEM IC50 in both WT and ELP5^{-/-} cells transfected with either EV (control) or hnRNPQ Um mutant, respectively. In Figure 6m, Western blot showed the expression of P53, BCL-2, BAX, and cleaved-CASP3 in both WT and ELP5^{-/-} cells transfected with EV or Flag-hnRNPQ Um in the presence and absence of GEM. We, therefore, speculate that the reviewer's suggestion refers most likely to Fig. 6m. As suggested, we added in **Fig. 6m** the phosphorylation status of P53 (at Ser46). And we found the expression of P53 and Ser46-phosphorylated P53 were rescued in ELP5^{-/-} cells transfected with Flag-hnRNPQ Um mutant under gemcitabine treatment; the expression of P53 and Ser46-phosphorylated P53 were also enhanced in WT cells transfected with Flag-hnRNPQ Um under gemcitabine treatment.

#11 figure 7. Similar analysis should be performed with other U34-tRNA modification enzymes (all ELP subunits, CTU1/2 etc). Is this only specific to ELP5?

--RE: We performed IHC analysis of all Elongator subunits (ELP1-4,6), CTU1, CTU2, ALKBH8, and P53, together with ELP5 and hnRNPQ results in the initial version in tissue microarray (Cohort 2), and found low expression of all ELPs, CTU1, CTU2, ALKBH8, hnRNPQ, and P53 were associated with poor survival outcomes in GBC patients (**New Fig. 7b, New Supplementary Fig. 9b**). Therefore, the poor survival outcomes in GBC patients was not particularly specific to ELP5. We also assessed the correlation between all U₃₄ tRNA-modifying enzymes and hnRNPQ, P53 expression via histoscore in IHC. As shown in **New Supplementary Fig. 9c**, all U₃₄ tRNA-modifying enzymes, hnRNPQ and P53 expression in GBC patients were positively correlated with each other.

The p53 status of the patient analyzed should be provided. The impact of p53 mutations (also seen in GBC) in this analysis should be further discussed/clarify in the manuscript.

--RE: While we provided the P53 expression in tissue microarray by IHC (New Fig. 7b), we now added new data to show P53 mutational status of Cohort 2, as well as the impact of P53 mutation (New Supplementary Fig. 9a, New Supplementary Table 7). GBC patients with P53 mutation exhibited poorer survival outcomes than wild type P53 (New Supplementary Fig. 9a). Therefore, both lower P53 expression (New Fig. 7b) and P53 mutational status (New Supplementary Fig. 9a) in GBC patients were correlated with poorer survival outcomes. We further discussed this point in the Discussion section (lines 427-433).

#12 Authors must add the reference to the recent paper describing that elongator regulates the IRES-dependent translation of LEF1 through codon-specific translation regulation of the ITAF protein DEK by Delaunay et al 2016. The present work is greatly inspired by this paper (and the model herein), which is curiously not mentioned at all. This should also be added in the discussion.

--RE: We greatly appreciate the reviewer's suggestion of adding the important reference (Delaunay S, et al. *J Exp Med*, 2016, 213, 2503-2523.). Indeed, this paper, along with two other papers (Rapino F, et al. *Nature*, 2018, 558, 605-609; Close P, et al. *J Biol Chem*, 2012, 287, 32535-32545) inspired us for exploring the mechanism of ELP5 in GEM resistance. Now we added this reference (Delaunay S, et al. *J Exp Med*, 2016, 213, 2503-2523.) in Ref. 33 and incorporated it into the Discussion section (lines 434-435). The other two papers have already been cited as Ref. 44 and Ref. 27, respectively.

#13 Line 118. It is not clear what the author want to claim here. Can we compare GEM and CISPLATIN? Do the author imply that cisplatin resistance occurs through a similar mechanism? If this is the case, this should be assessed experimentally.

--RE: We performed experiments to assess P53 expression and activation under the condition of cisplatin treatment, as we did under GEM treatment. As shown in New Supplementary Fig. 6a, total and Ser46-phosphorylated P53 were significantly down-regulated in ELP5-depleted GBC cells under cisplatin treatment, highly similar to those of GEM treatment. Down-regulated proteins also include P21 and pro-apoptotic protein BAX, suggesting that loss of ELP5 contributes to cisplatin resistance

through P53 regulation as well.

Additional comments

#1). Figure 2K is unreadable. Image should be made more clear and a quantification should be provided.

--RE: As suggested, in Fig. 2k, we have replaced the original immunofluorescence images of KI-67 (upper panel) and TUNEL assay (lower panel), with higher exposure and larger magnification pictures, and added bar graphs for the quantitative histogram (right panel).

Furthermore, in Supplementary Fig. 3h, we also replaced the original immunohistochemical pictures (KI-67 and Tunnel staining in GBC-SD xenograft models) with higher exposure and larger magnification pictures, and added bar-graphs for quantitative histogram.

#2). Does the expression of ELPs and U34-tRNA modification enzymes change in response to GEM?

--RE: As suggested, we now added in a New Supplementary Fig. 4i-k the data of all Elongator subunits (ELP1-6), CTU1, CTU2, ALKBH8, hnRNPQ, and P53 protein and mRNA expression, and thiolated tRNA (tE^{UUC}) abundance with or without GEM treatment. We found all Elongator subunits (ELP1-6), as well as hnRNPQ, P53, and pSer-P53, were significantly up-regulated at protein levels in the presence of GEM treatment (New Supplementary Fig. 4i), although the expression of only certain elongation factors (ELP2, ELP3, ELP5) and P53 was up-regulated at mRNA levels (New Supplementary Fig. 4j). The expression of the following three proteins did not respond to GEM: CTU1, CTU2, and ALKBH8 (New Supplementary Fig. 4i & 4j). What's more, the abundance of thiolated tRNA was also increased under GEM treatment (New Supplementary Fig. 4k). We have incorporated these points in the Discussion section (lines 404-413).

#3). p53 is one target of hnRNPQ, it would be good to know if hnRNPQ targets expression of other genes that depend on IRES for translation such as VEGF, MYC, cIAP, and others. Authors should test the expression of other potential targets of hnRNPQ. This should also be discussed as such by the authors, who also may be interested in exploiting the possibility that besides p53 mRNA, hnRNPQ may regulate translation of a number of mRNAs encoding proteins involved in tumor resistance to GEM

--RE: According to the reviewer's suggestion, we first searched the literature to look for the potential hnRNPQ-regulated genes. Indeed, in a paper published in *Cell Death Dis* (Lai CH, et al. *Cell Death Dis*, 2017, 8, e255.), Lai CH, et al. performed RNA-immunoprecipitation assay followed by next-generation sequencing (NGS) to identify hnRNPQ binding mRNAs, and filtered out a group of genes that were potentially regulated by hnRNPQ in cap-dependent and -independent manner. We carefully analyzed their NGS data and found that hnRNPQ protein could pull down transcripts of RUNX3, PTEN and DCK, all of which are tumor resistant genes, the loss of their functions are associated with GEM resistance in various cancer types (*Pancreatic cancer: Horiguchi S, et al. Molecular oncology*, 2013, 7, 840-849 and *Gu J, et al. Cancer Lett*, 2016, 380, 434-441; *gallbladder cancer: Nakano T, et al. Biochem Biophys Res Commun*, 2015, 464, 1084-1089). AURKA, a protein that has not yet shown to be related to GEM resistance, was also reported to be regulated by hnRNPQ in a cap-dependent and -independent translation manner (Lai CH, et al. *Cell Death Dis*, 2017, 8, e2555). Therefore, instead of *VEGF*, *MYC*, and *cIAP* the reviewer suggested, we evaluated the effects of hnRNPQ on the expression levels of RUNX3, PTEN, DCK, and AURKA. We found that the protein expression of all these genes was down-regulated in the hnRNPQ-knock-down cells. However, the mRNA expression of these genes was not altered (New Supplementary Fig. 8i & 8j; i, for protein and j, for mRNA). These data indicated that hnRNPQ indeed regulated the translation of a number of mRNA-encoding proteins that are involved in tumor resistance to GEM. We have incorporated these points into the Discussion section (lines 452-459).

#4). *In suppl table 4, other p53 ITAFs proteins appear to display a very high frequency of U34 modification codons. Therefore, they could represent ELP5 translational targets and could in one way or another contribute to the observed phenotype. Authors should test the expression of other ITAFs. This should also be further discussed as such by the authors in the manuscript.*

--RE: The Supplementary Table 4 in the original manuscript is now Supplementary Table 5 in the revised version. To address the reviewer's suggestion, we added a visual histogram of U₃₄ cognate codons percentage in New Fig. 6a. We also added the protein expression of other ITAFs protein in New Fig. 6b. We found that, while hnRNPQ was drastically reduced, other ITAFs were either modestly reduced (DAP5, PSF, ANXA2) or barely altered (RHA, RPL26, TCP80, and hnRNPL) in ELP5-depleted GBC cells. We added these points in the Discussion section (line 459-462).

#5). *The manuscript should be strongly edited. English should be improved throughout.*

--RE: As suggested, the English have been improved by an expert whose first language is English.

Reviewer #3 (Remarks to the Author): Expert in biliary tract cancers

The authors present data from a CRISPR screen in NOZ cells treated with gemcitabine, and identify ELP5 disruption as the main driver of gemcitabine resistance. The findings are validated in one additional cell line with different probes. Biological effects of loss of ELP5 are nicely described in the manuscript. Functional interaction of ELP5 with P53 and hnPRNPQ is also shown. Finally, clinical implications of ELP5 are demonstrated in human tissues and human PDX.

Data are novel and comprehensive. Data are interesting and well presented. However, there are some main concerns that should be addressed, please see below.

--RE: We greatly appreciate the reviewer's positive comments as well as suggested experiments.

1. Authors identified 210 hits that were present in the gemcitabine resistant cells. The top three were represented by DCK, P53 and ELP5. It would be useful to have information on which kind of disruption were identified in these genes from their NGS. Authors should also present the baseline mutational pattern of these cells to correctly interpret the data of the screen. Are there any P53 mutations in the NOZ cells?

--RE: As suggested, we now presented in Supplementary Fig. 1b & 1c the disruption of 210 gene hits by Gene Ontology (GO) and Kyoto Encyclopedia of Genes and Genomes (KEGG) pathway analysis and provided in Supplementary Table 2 & 3 all results of GO and KEGG analysis. We also provided 210 gene hits list in Supplementary Table 1.

To test if there is any P53 mutation as the reviewer asked, we amplified the exons of P53 by PCR in both NOZ and GBC-SD cells, and sequenced the entire exons. We found that there is no mutation in P53 isolated from either cells (**Data not shown**). We described the P53 genotype status in **Methods- Cell culture and reagents** (lines 493-494).

2. The doses used for gemcitabine look very high. Even the sensitive cells (i.e. Fig 2B sg-vector control cells) have an IC50 of 10uM, which is extremely high compared to the literature where IC50 for GEM is usually in the range of nM (Lampis Gastroenterology 2018, Sekine Anticancer research 2018).

--RE: The gemcitabine we used was purchased from Eli Lilly (USA) (trade name: GEMZAR), as the same source of *Lampis et al.*, while in the report by *Sekine et al.* the gemcitabine was obtained from Sigma-Aldrich (USA). That is the only difference in terms of the drug we used vs *Sekine et al.* used.

We have tested the drug resistance of four GBC cell lines, including NOZ, GBC-SD, SGC-996 and EH-GB1, at GEM concentration from nM to M range and calculated IC₅₀ to GEM. We found that NOZ is the most sensitive cells to GEM (IC₅₀ = 8.5 μM), followed by GBC-SD (IC₅₀ = 20.4 μM) (**New Supplementary Fig. 1a**). We selected NOZ and GBC-SD cells for all of our experiments, based on the IC₅₀ data from our drug screen experiments (**New Supplement Fig. 1a**). These two cell lines are the most frequently used cell lines in GBC basic researches (*Li M, et al. Nat Genet, 2014, 46, 872-876; Li M, et al. Gut, 2019, 68:1024-1033.*).

Actually, the IC₅₀ results we had are quite consistent to that reported in the literature (*Yu J, et al. Oncol Lett, 2018, 15:3305-3312; Wang, et al. Cell Death Dis, 2017, 8:e2770; Li Y, et al. Cell Biochem Biophys, 2014, 70:1337-1342; Makiyama A, et al. Anticancer Drugs, 2009, 20:123-130.*). In these publications, IC₅₀ of NOZ and/or GBC-SD cell lines for GEM are all in μM range. Regarding the two publications you mentioned that exhibited higher sensitivity than our observation, we have examined their results carefully. In the paper by *Lampis A, et al. (Gastroenterology, 2018, 154:1066-1079)*, the cells used are cholangiocarcinoma cell lines. Although both cholangiocarcinoma and gallbladder cancer belong to bile duct cancer, these two cancer types exhibit very different biological and genetic characterization. Therefore, it is not surprising to see their differences in their resistance to GEM. As for the article by *Sekine et al. (Anticancer Research, 2012. 32(8):3213-3218)*, we were unable to find the above-mentioned paper published in 2018. We assume that the reviewer had a typo, mistaken 2018 for 2012. In this article, the calculated IC₅₀ of NOZ to GEM is 0.66 μM, smaller than our calculated number (8.5 μM), but still somehow in the μM range. Therefore, our results are in agreement with that in the literature.

3. Which dose of GEM was used in the screening, and what was considered residual cells (i.e. less than 20% from original)?

--RE: The dose of GEM used in our screening is 10 μM (for 14 days). At this concentration and duration, **less than 10%** of the sgRNA library-infected NOZ^{Cas9} cells were able to survive. However, none of the

control NOZ^{Cas9} cells infected with non-specific sgRNA could survive (lethal treatment) at this concentration and duration. These ~10% surviving cells were harvested for extraction of genomic DNA, which were used for amplifying infected sgRNA sequences by PCR and subsequently next-generation sequencing. We described this in **Methods- Pooled CRISPR screen under gemcitabine treatment** (lines 532-534).

4. Conversely, GEM seems surprisingly effective in the in vivo experiments, especially considering the low doses (50mg/kg vs 100-150 mg/kg that is usually used). Surprisingly they have quite resistant cells in vitro (IC50 in the uM range, much higher than usually used), but very sensitive xenograft models (lower doses for in vivo exp). How would authors justify this discrepancy? Did they use different forms of gemcitabine?

--RE: For *in vivo* and *in vitro* experiments, we used the exactly same form of gemcitabine, obtained from Eli Lilly (USA) (trade name: GEMZAR), which was widely used in clinic and in basic researches (Lampis A, et al. *Gastroenterology*, 2018, 154, 1066-1079; Luo K, et al. *EMBO J*, 2017, 36, 1434-1446; Hill R, et al. *Cell Death Dis*, 2013, 4, e791).

The dose and frequency of gemcitabine intraperitoneal injection were done according to literature (Shukla SK, et al. *Cancer cell*, 2017, 32, 71-87; Al-Ejeh F, et al. *Clin Cancer Res*, 2014, 20, 3187-3197). In most literature, 100 mg/kg or a higher dose of GEM was used in 6 weeks or even 10 weeks old mice (Geller LT, et al. *Science*, 2017, 357, 1156-1160; Halbrook CJ, et al. *Cell Metab*, 2019, 29, 1390-1399). In the article by Al-Ejeh F, et al. (*Clin Cancer Res*, 2014, 20, 3187-3197), 50 mg/kg or 100 mg/kg were shown to exhibit similar inhibitory effects in the xenograft model.

In our study, to achieve a stable and more appropriate xenograft formation rate, we choose 4-week old athymic nude mice. In our pre-experiments, we found that the 4-week old athymic nude mice were unable to tolerance a regiment of gemcitabine at 100 mg/kg or higher every three days because of severe side effects even lethal. Therefore, we use 50 mg/kg for *in vivo* experiment. As for higher sensitivity *in vivo*, we speculate that the specific microenvironment *in vivo* (unknown specific cellular factors) may promote or sensitize the inhibitory effects of gemcitabine. We are interested in identifying the specific factors in the microenvironment in our future study.

5. Authors show that ELP5 deletion inhibits ELP3 and ELP4 proteins levels. What about the other components ELP1 ELP2 and ELP6? They should present the data.

--RE: We did additional experiments and had data to show the expression levels (at both mRNA and protein levels) of all ELPs, including ELP1, ELP2, ELP3, ELP4, ELP5, and ELP6 in ELP5-depleted and control cells (see New Fig. 3a and New Supplementary Fig. 4b). Our data showed that the levels of protein, but not mRNA, of ELP1, ELP2, and ELP6 were also down-regulated in ELP5-depleted GBC cells.

6. In figure 5A: ELP5 increases under gemcitabine exposure. How do authors justify this based on the experiments where they showed that ELP5 reduction is a mechanism adopted to become resistant?

--RE: We assessed the ELP1 - ELP6 expression and modified tRNA abundance under gemcitabine treatment. We found that ELP1 - ELP6 were significantly up-regulated under gemcitabine treatment, and Elongator's function activity (tRNA modification) was also enhanced under gemcitabine (New Supplementary Fig. 4i-k). A previous study confirmed that gemcitabine had a DNA methyltransferases inhibitor activity (*Gray SG, et al. Int J Mol Med, 2012, 30, 1505-1511*), consistent with finding that ELP5 mRNA level was increased under gemcitabine treatment (New Supplementary Fig. 4j). We analyzed the mRNA and DNA methylation levels of ELP5 in Cancer Cell Line Encyclopedia (CCLE) database, we found ELP5 mRNA level was the lowest (**Fig 1a in Point by point responses**) but DNA methylation rate in the promoter was the highest in various cancer types (**Fig 1b in Point by point responses**). Thus, we speculate that gemcitabine could demethylate the hypermethylated promoter of ELP5 genome to activate ELP5 transcription and protein expression (we have discussed this on Discussion, lines 404-413). The up-regulated Elongator complex could prime the Elongator/hnRNPQ/P53 axis to promote gemcitabine-induced cytotoxic effects.

Here is our justification for our results: in ELP5-knockout GBC cells, the integrity and stability of the Elongator complex were impaired and P53 expression was inhibited, resulting in higher resistance to GEM. In ELP5-expressing WT GBC cells, GEM treatments led to an increased expression of ELP5, which may further sensitize the cells to the toxicity of GEM, resulting in an increased sensitivity of the cells to the GEM. This data is in consistence with the results from our ELP5-overexpressing experiments, in which overexpression of ELP5 in control GBC cells (transfected with non-specific sgRNA) could promote gemcitabine-induced apoptosis (New Fig. 5h & 5i, lanes 1-2).

Fig. 1 ELP5 expression across the Cancer Cell Line Encyclopedia (CCLE) cancer cell lines. **a** The ELP5 mRNA levels across CCLE cancer cell lines with bile duct cell lines in red. **b** The ELP5 genomic DNA methylation frequency across the CCLE cancer cell lines with bile duct cell lines in red.

7. From figure 5H and 5I it looks that the main determinant of resistant is P53. Indeed reduction of P53 caused resistant also in WT, independently on ELP5. I believe these data reduces the strength of their hypothesis on ELP5 driving role, especially given that P53 resulted as a main hit from their screening.

--RE: We apology for the poor description of our data that caused the confusion and misunderstanding.

In order to better describe what we really wanted say, we generated ELP5-overexpressing GBC cells that were transfected with either P53-sgRNA (P53_sg1, or P53-knockout) or non-specific control sgRNA (sgNC, or P53^{WT}). As shown in **New Fig. 5h & 5i**, only overexpression of ELP5 in P53^{WT}-GBC cells, but not in P53-knockout cells, could enhance gemcitabine-induced apoptosis. These data demonstrate that ELP5-mediated gemcitabine sensitivity is dependent on P53 expression. Or in another word, ELP5 is at upstream of P53 in the signaling cascade.

8. I am not sure about the relevance of the mini-PDX, because it looks like the tumours do not grow in the mice, and are explanted after just 7 days.

--RE: Mini-PDX model is a modified PDX model with the primary tumor cells seeded in a capsule and then subcutaneously implanted to nude mice, followed by intraperitoneal injection of chemotherapeutic agents for 7 days, see in **Methods- Mini-patient derived xenograft (mini-PDX) model (line 641)**.

Mini-PDX model has advantages of being of higher engraftment rate, more time-saving, and more economical, and of high experimental consistency when compared to those of the traditional PDX models (Zhang F, et al. *Cancer communications*, 2018, 38, 60). With the mini-PDX models, we were able to consistently obtain quantitative results of chemotherapeutic agents for every model representing

specific patients, allowing easy and reliable comparison of the sensitivity or resistance between different tested agents. We use this model to screen and identify *in vivo* potential sensitive drug for personalized treatment of GBC patients in our clinic routines (Zhan M, et al. *Cancer communications*, 2018, 38, 48.). In the present study, we utilized the quantitative results of gemcitabine response in mini-PDX models, to correlate the expression levels of ELP5/hnRNPQ/P53 and gemcitabine sensitivity in GBC patients, respectively (New Fig. 7a). The results showed that lower level of ELP5, hnRNPQ and P53 expression was associated with poor sensitivity to gemcitabine in mini-PDX models.

Minor:

1). The manuscript needs an overall review of the English language, especially in the abstract and the introduction.

--RE: Per your suggestion, the writing have been improved. Many thanks.

2. Page 4 line 101, please specify is it is the ELP5 mRNA expression that is associated to IC50.

RE: As suggested, we modified this with “the ELP5 mRNA expression” in line 104.

REVIEWERS' COMMENTS:

Reviewer #1 (Remarks to the Author):

The manuscript has been improved by the modifications. Of note, there are independent validations that the screen is indeed a bonafide one. For example, the supplementary data identifies metabolic pathways and associated genes, including GLUT1 and TKT as potential regulators of gemcitabine response. The authors should cite Shukla et al Cancer Cell 2017 (PMID: 28898700) that also identified both of these genes and associated pathways as potential regulators of therapy response in human patients. It is understandable that the authors chose to use the most altered pathways; the identified pathways may still impart metabolic alterations to induce resistance to gemcitabine.

Reviewer #2 (Remarks to the Author):

I wish to thank and congratulate the authors for their answers and the extensive experimental work they provided. Most of my concerns have been correctly addressed.
Best wishes

Reviewer #3 (Remarks to the Author):

Authors have addressed the issues raised and they added a remarkable amount of details and experiments to the work.

Point by point response to the reviewers' comments

Reviewer #1 (Remarks to the Author):

The manuscript has been improved by the modifications. Of note, there are independent validations that the screen is indeed a bonafide one. For example, the supplementary data identifies metabolic pathways and associated genes, including GLUT1 and TKT as potential regulators of gemcitabine response. The authors should cite Shukla et al Cancer Cell 2017 (PMID: 28898700) that also identified both of these genes and associated pathways as potential regulators of therapy response in human patients. It is understandable that the authors chose to use the most altered pathways; the identified pathways may still impart metabolic alterations to induce resistance to gemcitabine.

--RE: Thanks for reviewer's positive review. As suggested, we revised our results description about the identified genes (lines 96-99) and cited this paper (Shukla SK, et al. Cancer Cell, 2017, 32(3):71-87.e7.) in **Ref. 20**.

Reviewer #2 (Remarks to the Author):

I wish to thank and congratulate the authors for their answers and the extensive experimental work they provided. Most of my concerns have been correctly addressed.

Best wishes

--RE: Thanks. No response to Reviewer #2 is needed.

Reviewer #3 (Remarks to the Author):

Authors have addressed the issues raised and they added a remarkable amount of details and experiments to the work.

--RE: Thanks. No response to Reviewer #3 is needed.